# Locality-based 3-D multiple-point statistics reconstruction using 2-D geological cross-sections

Qiyu Chen [1,2,3], Gregoire Mariethoz [2], Gang Liu [1,3,*], Alessandro Comunian [4], Xiaogang Ma [5]

[1] School of Computer Science, China University of Geosciences, Wuhan 430074, China
[2] Institute of Earth Surface Dynamics, University of Lausanne, 1015 Lausanne, Switzerland
[3] Hubei Key Laboratory of Intelligent Geo-Information Processing, China University of Geosciences, Wuhan 430074, China
[4] Dipartimento di Scienze della Terra "A.Desio", Università degli Studi di Milano, Milan, Italy
[5] Department of Computer Science, University of Idaho, 875 Perimeter Drive MS 1010, Moscow, ID 83844-1010, USA

*Correspondence to: liugang67@163.com (G. Liu)*

**Abstract:** Multiple-point statistics (MPS) has shown promise in representing complicated subsurface structures. For a practical three-dimensional (3-D) application, however, one of the critical issues is the difficulty to obtain a credible 3-D training image. However, bidimensional (2-D) training images are often available because established workflows exist to derive 2-D sections from scattered boreholes and/or other samples. In this work, we propose a locality-based MPS approach to reconstruct 3-D geological models on the basis of such 2-D cross-sections (3DRCSs), making 3-D training images unnecessary. Only several local training sub-sections closer to the central uninformed node are used in the MPS simulation. The main advantages of this partitioned search strategy are the high computational efficiency and a relaxation of the stationarity assumption. We embed this strategy into a standard MPS framework. Two probability aggregation formulas and their combinations are used to assemble the probability density functions (pdfs) from different sub-sections. Moreover, a novel strategy is adopted to capture more stable pdfs, where the distances between patterns and flexible neighborhoods are integrated on several multiple grids. A series of sensitivity analyses demonstrate the stability of the proposed approach. Several hydrogeological 3-D application examples illustrate the applicability of the approach 3DRCSs in reproducing complex geological features. The results, in comparison with previous MPS methods, show better performance in portraying anisotropy characteristics and in CPU cost.

**Keywords:** Three-dimensional reconstruction, Multiple-point statistics, Locality, Cross-sections, Non-stationarity, Probability aggregation

## 1. Introduction

3-D characterization of geological architectures plays a crucial role in the quantification of subsurface water, oil and ore resources (*Chen et al.*, 2017, 2018; *Foged et al.*, 2014; *Hoffman and Caers*, 2007; *Jackson et al.*, 2015; *Kessler et al.*, 2013; *Raiber et al.*, 2012; *Wambeke and Benndorf*, 2016). Heterogeneity and connectivity of sedimentary reservoirs exert controls

on underground fluid transport (*Gaud et al.*, 2004; *Renard and Allard*, 2013; *Weissmann et al.*, 1999) which is vital to quantify and forecast the formation and distribution of subsurface resources. For a practical 3-D application, however, these attributes are notoriously difficult to characterize and model since the informed data we can acquire are very sparse. Two-point geostatistics (*Goovaerts*, 1998; *Pyrcz and Deutsch*, 2014; *Ritzi*, 2000) and object-based methods (*Deutsch and Tran*, 2002; *Maharaja*, 2008; *Pyrcz et al.*, 2009) are difficult to reproduce anisotropic features and connectivity patterns properly (*Heinz et al.*, 2003; *Klise et al.*, 2009; *Knudby and Carrera*, 2005; *Vassena et al.*, 2010) due to the lack of high-order statistics and the difficulty in parameterization. To overcome the abovementioned limitations, multiple-point statistics (MPS) was developed over the recent years and has shown prospects in modeling subsurface anisotropic structures, such as porous media, hydrofacies, reservoir, and other sedimentary structures (*Dell Arciprete et al.*, 2012; *Hajizadeh et al.*, 2011; *Hu and Chugunova*, 2008; *Oriani et al.*, 2014; *Pirot et al.*, 2015; *Wu et al.*, 2006).

A first MPS approach was suggested by *Guardiano and Srivastava* (1993) which is designed to reproduce heterogeneous geometries by extracting spatial patterns from training images directly rather than through variograms. A training image is a conceptual model derived from observations, and it bears a crucial role in MPS-based stochastic simulation. The first efficient implementation of MPS was developed by *Strebelle* (2002) on the basis of a tree structure. Several attempts have thereafter focused on improving MPS algorithms (*Arpat and Caers*, 2007; *Caers*, 2001; *Mariethoz et al.*, 2010; *Straubhaar et al.*, 2011; *Tahmasebi et al.*, 2012; *Wu et al.*, 2008; *Yang et al.*, 2016; *Zhang et al.*, 2006). With these methods, training images are scanned with a fixed search template and the MPS patterns are stored in a tree or a list data structure. An important difficulty lies in choosing the size of data template to best reproduce large-scale structures (*Strebelle*, 2002). The larger the size of the data event, the fewer replicates of this data event will be found over the training images for inferring the corresponding conditional probability density function (cpdf). However, when the size of data template is too small, large scale structures of the training image cannot be reproduced (*Mariethoz et al.*, 2010). In addition, a search template including too many nodes can lead to storing a large number of patterns, increasing CPU cost and memory consumption. The multiple-grids concept (*Tran*, 1994; *Strebelle*, 2002) mitigates the above-mentioned limitations, but they still present due to the rigidity of data templates and multiple grids. A more straightforward MPS method, Direct Sampling (DS), was proposed by *Mariethoz et al*. (2010) where the high order statistics are sampled directly from the training image without storing patterns and without the need of multiple grids. One of the main advantages of this approach is that several types of distances between patterns can be considered, making it possible to simulate continuous variables, or even multivariate simulation. As an approximation, pattern distance was used to express the matching degree between the neighborhood of a node and a data event in the training image (*Chugunova and Hu*, 2008; *Mariethoz et al.*, 2010, 2015).

No matter which MPS algorithm is used, a suitable training image is the fundamental requirement. Although such algorithms are gaining popularity in hydrogeological applications (*Hermans et al.*, 2015; *He et al.*, 2014; *Høyer et al.*, 2017; *Hu and Chugunova*, 2008; *Huysmans et al.*, 2014; *Jha et al.*, 2014; *Mahmud et al.*, 2015), they still suffer from one vital limitation: the lack of training images, especially for 3-D situations. Object-based or process-based techniques are one possibility to build 3-D training images (*de Marsily et al.*, 2005; *de Vries et al.*, 2009; *Feyen and Caers*, 2004; *Maharaja*,

2008; *Pyrcz et al.*, 2009). Besides inherent limitations in the parameterization of these algorithms, it is also challenging to reproduce the various aspects of geological geometries from a high-resolution outcrop map, or even from insufficient borehole data (*Comunian et al.*, 2014; *Pirot et al.*, 2015). To overcome this difficulty of obtaining 3-D training images, scholars have attempted to use low-dimensional data (e.g. boreholes, cross-sections, outcrop and remote sensing and geophysical images) to reconstruct 3-D models directly instead of a training image in the entire 3-D domain (*Bayer et al.*, 2011; *Comunian et al.*, 2011; *Hu et al.*, 2011; *Weissmann et al.*, 2015). In particular, a reconstruction method of partial data sets was proposed by *Mariethoz and Renard* (2010) by using and adapting the DS algorithm. However, large-scale 3-D models contain millions of nodes, thus a very large number of scan attempts will be carried out for each simulated node by using this method, especially in early stages of a simulation due to the sparse known data. Therefore, this method still suffers from a severe computational burden for fine 3-D applications. Moreover, it assumes stationarity of the modeled domain, which is not often the case in practice. *Comunian et al.* (2012) proposed an approach to tackle the lack of a full 3-D training image using sequential 2-D simulations with conditioning data (s2Dcd): a 3-D domain is filled by preserving an overall coherence due to that a series of 2-D simulations performed using 2-D training images along orthogonal directions. However, this strategy is difficult to characterize the connectivity of structures in all directions of a 3-D domain, because each 2-D simulation only considers the high-order statistics in this direction. Moreover, it also suffers from the limitation of nonstationarity of geological phenomena due to the global search in a 2-D plane. To integrate the benefits of the both approaches, *Gueting et al.* (2017) proposed a new combination of the two existing approaches. The combination is achieved by starting with the sequential two-dimensional approach (*Comunian et al.*, 2012), and then switching to the three-dimensional reconstruction approach (*Mariethoz and Renard*, 2010). However, the above-mentioned limitations of the two approaches still remain because this combination is an optimization of the workflow, and does not substantially improve the methods. To combine the cpdfs from different directions, several probability aggregation methods were tested and discussed (*Allard et al.*, 2012; *Bordley*, 1982; *Genest and Zidek*, 1986; *Journel*, 2002; *Krishnan*, 2008; *Mariethoz et al.*, 2009; Stone, 1961). Other 3-D applications to represent geological structures using MPS and partial data include filling in the shadow zone of a 3-D seismic cube (*Wu et al.*, 2008), generating small scale 3-D models of porous media (*Okabe and Blunt*, 2007) and building a 3-D training image with digital outcrop data (*Pickel et al.*, 2015).

From another perspective, using very common workflows, geologists can obtain 2-D geological maps or sections from scattered boreholes and/or other samples by studying analogs (*Caumon et al.*, 2009). With increasingly sophisticated data acquisition methods, 2-D high-resolution images can be acquired. For example, large-scale outcrop maps can be captured by using terrestrial lidar (*Dai et al.*, 2005; *Heinz et al.*, 2003; *Nichols et al.*, 2011; *Pickel et al.*, 2015; *Zappa et al.*, 2006), and fine-scale pore images can be derived from progressive imaging techniques (*Zhang et al.*, 2010). Therefore, there are many ways to acquire low-dimensional data for reconstructing a full 3-D model. In practice, however, these works using real geological analogs or sections as training images still face significant non-stationarity due to the heterogeneity of geological phenomena and processes (*Comunian*, 2011; *de Vries et al.*, 2009).

To address the insufficient access to a 3-D training image and the challenge of non-stationarity, we present a new strategy to reconstruct 3-D geological heterogeneities using 2-D cross-sections (3DRCSs) instead of an entire training image. Compared to previous MPS implementations relying on partial data, our proposal is to use only several local sub-sections closer to the simulated node as training images, rather than full planes perpendicular to the *x*, *y* and *z* directions (*Comunian et al.*, 2012) or searching in the entire 3-D domain (*Mariethoz and Renard*, 2010). Against to the filling by a series of 2-D simulations in s2Dcd (*Comunian et al.*, 2012), a random simulation path containing all uninformed locations is used so that MP statistics in a 3-D domain are captured. The local sub-sections are able to offer more coherent and reliable statistics since they are spatially closer to the simulated node which is going to be simulated. Moreover, the original cross-sections are divided into many sub-sections according to their spatial relationships, thus non-stationarity is reduced since it is restricted into a local cube consisting of six or fewer sub-sections. In principle, our proposal can be applied into any multiple-point stochastic simulation method. In this work, we embed this strategy into a standard MPS framework called ENESIM (*Guardiano and Srivastava*, 1993). The blocking strategy proposed in this work can significantly reduce the search space of training images, which makes it possible to get a 3-D reconstruction using ENESIM for a reasonable CPU cost. As with DS, in our approach MP statistics are not stored and the neighborhood is flexible. To integrate the patterns from different sub-sections, two probability aggregation formulas and their combinations are used. As an approximation of the matching degree between neighborhoods and data events, pattern distances are used to enhance the stability of cpdfs. Furthermore, we adapt multiple-grids into the approach 3DRCSs, where the geometries of data templates are not fixed for grids of different scales. Besides cross-sections, any other scattered samples can also be involved into the proposal as conditional data (hard data).

The remainder of this paper is organized as follows. Section 2 gives background information used in the following sections. Section 3 presents the main concepts of the locality-based 3-D MPS reconstruction using 2-D cross-sections and the detailed steps of the proposed approach. Section 4 shows a parameter sensitivity analysis and the performance comparison with other MPS algorithms. Section 5 gives a synthetic example in hydrogeology to illustrate the effectiveness of the approach 3DRCSs when facing the real geological field data. The final section discusses some concluding remarks and ideas for future work.

## 2. Background Information

### 2.1. Pattern Distance

A pattern distance $d\{\mathbf{N}_X, \mathbf{N}_Y\}$ is an approximation of the dissimilarity between patterns, which is used to compare the neighborhood of a node currently simulated with a data event in the training image (*Mariethoz et al.*, 2010). Approximate matches are accepted by using a distance threshold $t$. Namely for a data event $\mathbf{N}_X$ from the simulation grid, when the condition $d\{\mathbf{N}_X, \mathbf{N}_Y\} \leq t$ $(t \geq 0)$ is met, the pattern $\mathbf{N}_Y$ from the training image will be used to update the current cpdf. For a categorical variable, the distance can be formulated as:

$$d\{\mathbf{N}_X, \mathbf{N}_Y\} = \frac{1}{n}\sum_{i=1}^{n} a_i \in [0,1], \quad where \quad a_i = \begin{cases} 0 & if \ Z(x_i) = Z(y_i), \\ 1 & if \ Z(x_i) \neq Z(y_i). \end{cases} \tag{1}$$

For a non-stationary training image from an actual geological phenomenon, repeatability of spatial patterns could be weak so that it is hard to acquire a stable cpdf. Therefore, we adopt a patterns distance with a threshold as an approximation to sample more patterns and get a more stable cpdf.

## 2.2. Probability Aggregation

*Allard et al.* (2012) presented a comprehensive literature review for aggregating probability distributions. These can be divided into additive methods and multiplicative methods according to their mathematical properties. Linear Pooling formula (*Stone*, 1961) is a widely used method (for example, it was used by *Okabe and Blunt*, 2007) based on the addition of probabilities. It is appealing because of its flexibility and simplicity. Multiplicative methods include Bordley/Tau models and log-linear pooling (based on odd ratios) (*Bordley*, 1982; *Journel*, 2002; *Genest and Zidek*, 1986).

### 2.2.1. Linear Pooling Formula

The linear pooling formula, proposed by *Stone* (1961), probably is the most intuitive way of aggregating the probabilities $P_1, ..., P_n$ of an event $A$.

$$P_G(A) = \sum_{i=1}^{n} w_i P_i(A) \quad with \ w_1, ..., w_n \in \mathbf{R}^+. \tag{2}$$

In this formula, $w_i$ are positive weights and their sum must equal one to obtain a global probability $P_G \in [0,1]$.

### 2.2.2. Log-Linear Pooling Formula

The log-linear pooling formula is a linear operator of the logarithms of the probabilities (*Genest and Zidek*, 1986). If a prior probability $P_0(A)$ must be included, it is written as:

$$P_G(A) \propto P_0(A)^{1-\sum_{i=1}^{n} w_i} \prod_{i=1}^{n} P_i(A)^{w_i}. \tag{3}$$

$\sum_{i=0}^{n} w_i = 1$ is needed to verify external Bayesianity. There are no other constraints whatsoever on the weights $w_i$, $i = 0, ..., n$. The sum $S = \sum_{i=1}^{n} w_i$ plays an important role in this formula. If $S = 1$, the prior probability $P_0$ is filtered out because $w_0 = 0$. Otherwise, if $S > 1$, the prior probability has a negative weight and $P_G$ is further away from $P_0$ than

other probabilities. Conversely, if $S < 1$, $P_G$ is always closer to $P_0$. Therefore, we can adjust the influence of the prior probability $P_0$ on the aggregated result $P_G$ by changing the value of $S$.

## 2.3. Multidimensional Scaling and Kernel Smoothing

*Tan et al.* (2014) proposed a distance-based approach to evaluate the quality of MP simulation outcomes where the Jensen–Shannon (JS) divergence is used to depict the dissimilarity of MP histograms as a quantitative metric. The information in the dissimilarity of MP histograms can be visualized using multidimensional scaling (MDS) (*Caers*, 2011). MDS approximates these distances by a lower-dimensional Euclidean distance in Cartesian space, which facilitates the visualization of the dissimilarity of MP histograms.

*Hermans* et al. (2015) used an adaptive kernel smoothing (see *Park et al.*, 2013) to estimate the probability density of the data variable for each kind of realizations $f(\mathbf{Ref}^* | \mathbf{R}_i)$ in the $d$-dimension space inferred from MDS. This allows estimating the probability density distribution of the realizations around the reference. For each kind of realizations, its probability relative to the reference $P(\mathbf{R}_i | \mathbf{Ref})$ can be calculated by using Bayes' rule:

$$P(\mathbf{R}_i | \mathbf{Ref}) \approx P(\mathbf{R}_i | \mathbf{Ref}^*) = \frac{f(\mathbf{Ref}^* | \mathbf{R}_i)P(\mathbf{R}_i)}{\sum_{i=1}^{N} f(\mathbf{Ref}^* | \mathbf{R}_i)P(\mathbf{R}_i)}. \tag{4}$$

## 3. Methodology

### 3.1. Local Search Strategy of 3-D MPS Reconstruction

In the above-mentioned MPS methods using partial data whether searching an entire 3-D domain or complete sections, any locations of the training images are scanned even they are far away from the simulated node so that one spatial pattern will be carried to a distant position. Therefore, the use of these methods is restricted to stationary training images, which are in practice seldom available. In this work, we propose a local search strategy that allows palliating this problem, by taking into account the spatial relationships of the real geological cross-sections in a given 3-D domain.

As illustrated in Figure 1, a 3-D domain is segmented into nine small blocks by six cross-sections from three orthogonal directions where there are two sections in each direction. Every local block is surrounded by $n$ local sub-sections ($1 \leq n \leq 6$). It should be noted that, sometimes, local blocks are not closed (i.e. the surrounding sub-sections are less than six) (Figure 1b); and it is also allowed sections along some planes are missing; however, at least one section should be provided. For each unknown node in the local block (e.g. the gray cubes in Figure 1c), the MP statistics are captured from the surrounding sub-sections rather than from the entire sections. Namely, there are $n$ corresponding training images for each simulated node. These local sub-sections are the parts of the global cross-sections which are closer to the uninformed

nodes in the local block, thus they are more likely to be regarded as statistically representative. Data events are selected from the informed nodes (hard data) on three planes parallel to the sub-sections and through the current simulated node in three orthogonal directions by using a flexible neighborhood.

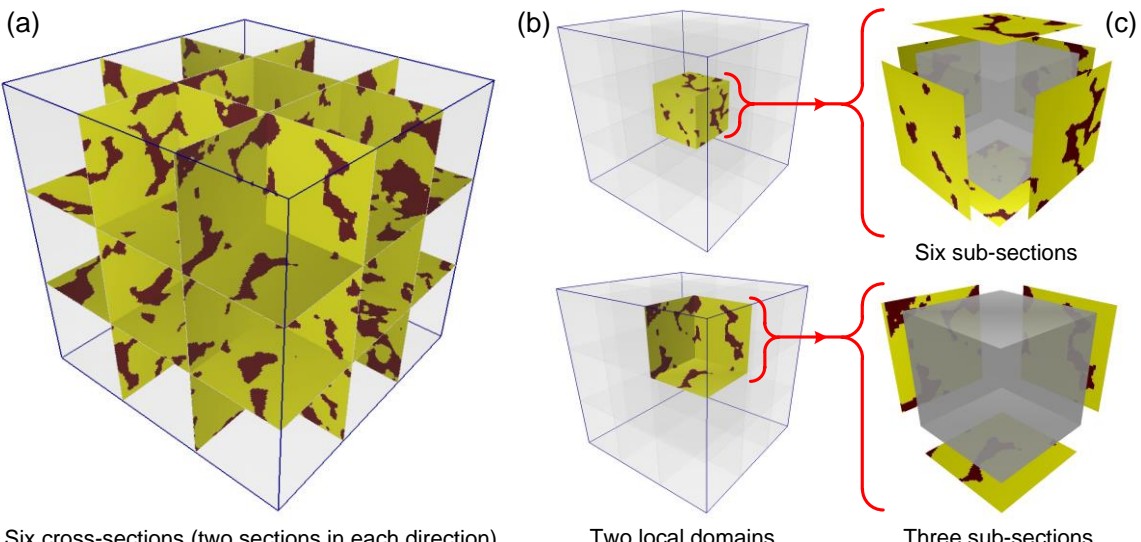

Six cross-sections (two sections in each direction)   Two local domains   Three sub-sections

**Figure 1.** Local sub-sections divided by their spatial relationships and the corresponding training images. (a) Six cross-sections in a 3-D domain: two sections along each direction; (b) two local domains: a central cube and a corner cube; (c) corresponding sub-sections (training images).

Another important point is related to handling of the search window when scanning a sub-section. Here, we allow all locations of a sub-section to be visited by the central node of a data event. The neighbor nodes of the data event can be placed in other adjacent sub-sections when matching with the training images. As shown in Figure 2, the area inside the blue line is the search window. If only the nodes of the data event are from the sub-section itself (case 1 on the figure), the training patterns are seriously reduced. We adopt a search strategy where neighbor nodes can be searched in the neighboring sub-sections (case 2 on the figure). Its main advantages are the coherence of the spatial patterns in a realization and the larger number of training patterns available. In addition, the size of the data events is constrained by the boundary of the global section, as illustrated in *Mariethoz et al.* (2010).

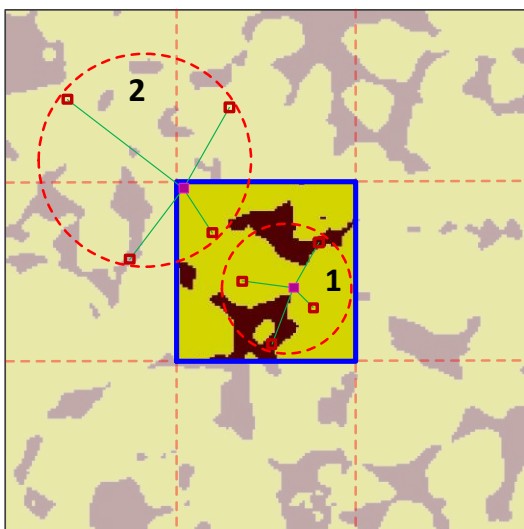

**Figure 2.** Search window in sub-sections.

If more cross-sections are available, a finer spatial subdivision can be used. In this case, the size of each sub-section is smaller and the computational cost is reduced significantly. However, extremely small training images cannot offer enough spatial patterns, thus a minimal sub-section size has to be considered. In practice, if there are many sections in each direction, a feasible solution is to select several ones as the references and others are used as conditioning data only.

### 3.2. Strategy for Aggregating the pdfs from Local Sub-Sections

As an additive aggregation method, the linear pooling formula corresponds to a mixture model, which is related to the union of events and to the logical operator OR (*Allard et al.*, 2012). This method is thus used to unite several independent probabilities into a global term $P_G$. The log-linear pooling formula, based on the multiplication of probabilities, is related to the intersection of events and to the logical operator AND. Therefore, we usually use such a method to aggregate the probabilities with significant correlation to acquire a conjunction probability.

In this study, $n$ pdfs ($1 \leq n \leq 6$) are computed from the surrounding local sub-sections (Figure 1). For the illustrative case proposed here, a local 3-D domain is surrounded by six sub-sections, and six pdfs are being aggregated. There are two parallel sub-sections (training images) in each direction. An additive aggregation operator is more appropriate to combine the two probability distributions from parallel sub-sections, since we just expect a larger number of samples and thus more robust pdf by uniting both. Then, three orthogonal pdfs are obtained. We then join these pdfs containing the statistics from different directions with obvious anisotropic features. This scenario needs a multiplicative method to combine the orthogonal pdfs so as to retain the features in all directions. In summary, an optimal probability aggregation strategy is proposed by the procedure described below:

1. Aggregate the pdfs collected along the same direction for parallel sub-sections using the linear pooling formula described in section 2.2.1.

2. Aggregate the orthogonal pdfs from the above step by using the log-linear pooling formula described in section 2.2.2.

Of course, the probability aggregation step is not required when for step 1 there is only one sub-section along a given plane, and for step 2 the pdf that along some direction are missing are simply not included in the aggregation process. For the step 1, the weights $w_1$ and $w_2$ are related to the distances between the current location and the two parallel sub-sections $d_1$ and $d_2$ s, and computed as:

$$w_1 = \frac{1/d_1}{1/d_1 + 1/d_2} \ , \ w_2 = \frac{1/d_2}{1/d_1 + 1/d_2} \ .$$

(5)

Such parameterization ensures that within-block trends are accounted for.

For the step 2, an influence of the prior probability is desired to tune the other orthogonal pdfs. Thus, we usually use $0 < w_0 < 1$, and set $w_i (i = 1, ..., n)$ equal, i.e. $w_i = (1 - w_0)/n$, where n is the number of pdfs to be aggregated. However, the weights $w_i (i = 1, ..., n)$ can also change, for example, they can vary at each simulation step as described in *Comunian et al.* (2012), according to the contributions of the different training images, while sum still respects the condition $\sum_{i=0}^{n} w_i = 1$.

### 3.3. Flexible Search Template on Multiple Grids

When large neighborhoods are considered, it is more difficult to find matching data events in the training image and thus a larger distance threshold $t$ is required to obtain a sufficient number of samples for an acceptable cpdf. This can lead to degrading small-scale features or the removal of categories that have a low proportion. To address this issue, we propose a novel implementation of multiple grids where the search template is flexible and the distance threshold $t$ varies according to the radius of the neighborhood.

As illustrated in Figure 3, an example of multiple grids with three levels is used to show the relationship between neighborhoods, search radius $R$ and distance threshold $t$ on different grids. A neighborhood is identified by the informed and/or simulated nodes located in the circle with a radius $R$ on the current grid and the current node (the gray nodes in Figure 3) as a central. The initial radius $R_0$ and distance threshold $t_0$ for the first grid are assigned as the input parameters. The radius $R$ linearly reduces to 1 from the first to the last grid, and the threshold $t$ similarly varies from 1 to 0. The neighboring nodes (hard data and previously simulated nodes) around the central node on the current grid are selected to build a data event according to the radius $R$ and the maximum number of points in the neighborhood. Therefore, a large data event is divided into several small parts placed on the different grids which results in smaller neighborhoods on each grid. An acceptable threshold $t$ is thus assigned to each neighborhood. For the last grid, the radius is reduced to 1 and at most there are eight nodes in a neighborhood. This strategy considers that small data events located on the last grid are much

more repetitive (thus easier to find) than the large data events of the first grid. Figure 3 shows the flexible use of multiple grids on one plan through the current node. In the local search strategy proposed in this work, three planes through the current simulated node in three orthogonal directions are considered to obtain the neighborhoods. Thus the same strategy will be applied on other two planes.

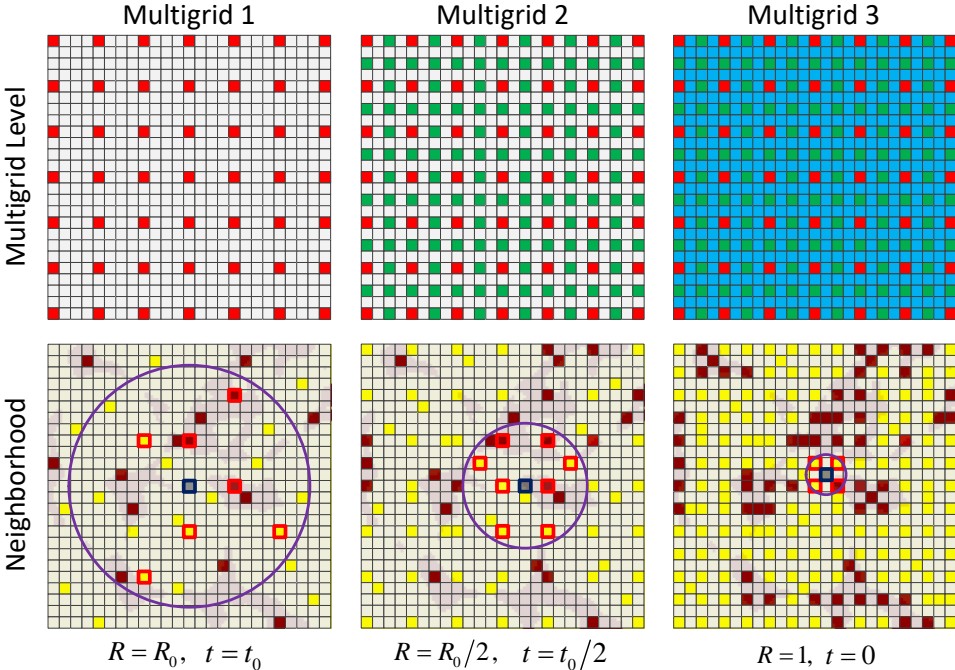

Multigrid 1      Multigrid 2      Multigrid 3

Multigrid Level

Neighborhood

$R = R_0, \quad t = t_0$      $R = R_0/2, \quad t = t_0/2$      $R = 1, \ t = 0$

**Figure 3:** An example of multiple grids and the corresponding neighborhoods, search radius $R$ and distance threshold $t$.

### 3.4. Step-by-Step Algorithm Using the Local Search Strategy

Based on the strategies proposed in the above sections, the detailed steps of our simulation algorithm proceed as illustrated in Algorithm 1.

---

**Algorithm 1:** Reconstruct 3-D geological structures using 2-D cross-sections

---

**1** Load data files, assign all points of the training data (cross-sections and other samples) into the SG.
**2** Record the indexes of the sections in $x$, $y$, $z$ directions and compute the prior proportions $\mathbf{P}_p$ of the local domains.

**3** **For** each multiple grid $g$ :

**4**      Define a random simulation path for grid $g$ according to the remaining nodes.

**5**      **Do** until all uninformed nodes in $g$ have been visited:

**6**          Get the index of current node $\mathbf{x}$, and identify its neighborhood $\mathbf{N}_x$.

**7**          Obtain the indexes of the closest sections around $\mathbf{x}$: $\{x_0, x_1\}$, $\{y_0, y_1\}$, $\{z_0, z_1\}$.

**8**          Randomly scan the sub-sections (TIs) and get the corresponding cpdfs (see algorithm 2).

---

| 9 | Get the prior proportion $\mathbf{p}$ of the local domain according to the location of the node $\mathbf{x}$. |
|---|---|
| 10 | Combine the cpdfs and $\mathbf{p}$ into a joint pdf using the strategy presented in section 3.2. |
| 11 | Randomly draw a value from the final pdf, and assign it to location $\mathbf{x}$. |
| 12 | **End** |
| **13** | **End** |

As mentioned above, we capture the MP statistics from several sub-sections of a local domain. Thus, the corresponding prior proportion should also be computed on the basis of these surrounding sub-sections (step 2). Comparing to s2Dcd, we use a fully random path on each multiple grid in the 3-D space and not within a specific section. For the current node, however, the MP statistics are only captured from several sub-sections in three orthogonal directions, because we only have 2-D cross-sections to scan and no a 3-D training image. Obviously, step 8 is the most important procedure in our simulation algorithm, and the idea is inspired from ENESIM (*Guardiano and Srivastava*, 1993) and DS (*Mariethoz et al.*, 2010). The main procedure is demonstrated in Algorithm 2.

---

**Algorithm 2:** Scan a local sub-section (training image) in one certain direction

**Input:** $\mathbf{x}$: current simulation location; $id$: index of the training image that will be scanned;

$\chi_0, \chi_1, \gamma_0, \gamma_1$: indexes of the closest training image s in the other two directions.

**Output: cpdf:** conditional probability density function from the current training image.

| 1 | **Function** $\text{ScanTI}(\mathbf{N}_x, id, \chi_0, \chi_1, \gamma_0, \gamma_1, \&\text{cpdf})$ |
|---|---|
| **2** | Get the sub-section $\text{sub\_}S$ (training image) according to the $id$ and $\chi_0, \chi_1, \gamma_0, \gamma_1$; |
| **3** | Set a random path $p$, and initialize the counter of matched patterns $sum = 0$; |
| **4** | **for** $i := 0 \rightarrow p.\text{size}()$ **such that** $i < p.\text{size}() \times f$ **do** |
| **5** | Sample a location in the training image and get the neighborhood $\mathbf{N_Y}$; |
| **6** | Compute the distance $d\{\mathbf{N_X}, \mathbf{N_Y}\}$ using equation (1) presented in section 2.1; |
| **7** | **if** $d\{\mathbf{N_X}, \mathbf{N_Y}\} \le t$ **then** |
| **8** | update the cpdf according to the facies of the central point in the training image; |
| **9** | $sum++$; |
| **10** | **end if** |
| **11** | **if** $sum > N_{\max}$ **then** |
| **12** | **break;** |
| **13** | **end if** |
| **14** | **end for** |
| **15** | **end Function** |

---

The fraction of the scanned training image $f$ and the distance threshold $t$ are borrowed from DS and they play the same roles. $\chi_0, \chi_1, \gamma_0, \gamma_1$ are the indexes of the closest training images in the other two directions and they are used to determine the current sub-section (training image). A new parameter, the maximum number of matched patterns from the training image $N_{max}$ is adopted to avoid unnecessary searches. For some small neighborhoods, especially in the last multiple grid, the cpdf will rapidly stabilize with the increasing number of matched patterns.

## 4. Parameterization and Performance Analysis

In this section, we apply 3DRCSs on several synthetic cases where the cross-sections are extracted from existing 3-D references. Using these examples, we perform a parameter sensitivity analysis and compare it with two widely used methods, DS-based 3-D reconstruction (*Mariethoz* and *Renard*, 2010) and s2Dcd (*Comunian et al.*, 2012). The workflows and algorithms proposed in this work are developed in the C++ programming language. All experiments presented in this paper are implemented on a laptop computer with Intel 4-Cores i5-62000U Quad-core CPU 2.30 GHz, 8 GB RAM and 64 bit Windows 10.

### 4.1. Parameter Sensitivity

The majority of parameters of 3DRCSs are similar to DS. Therefore, only the sensitivity of three parameters specific to 3DRCSs are tested against the 3-D reference shown in Figure 4, considering CPU cost and statistical and geometrical features of the realizations obtained. All cross-sections used in the following tests in the section 4.1 are extracted from this 3-D model.

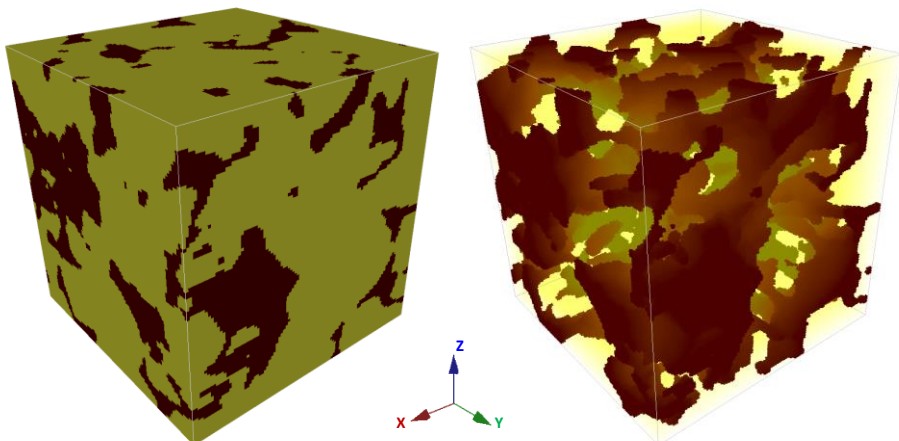

**Figure 4.** A sample of Berea sandstone from *Okabe and Blunt* (2007) is used as a 3-D reference ($100 \times 100 \times 100$ voxels). The crimson color represents pores and the yellow color represents matrix. The porosity of this model is 20.33%.

### 4.1.1. Number of Cross-Sections

The number of cross-sections $N_{cs}$ is a new parameter in the approach 3DRCSs. They are not only regarded as the training images and conditioning data, but also control the computing speed and the quality of the reconstructions. Figure 5 and Table 1 show different reconstructions and their statistical properties by increasing the sections in every direction. In this test, the number of cross-sections $N_{cs}$ in each direction increases from one to six, and other parameters are fixed: maximum search radius = 50, maximum number of points in a neighborhood = 35, distance threshold $t_0 = 0.2$, fraction of training image to scan $f = 0.8$, maximum of matched patterns from each training image = 100, number of multiple grids = 3, weights of the probability aggregation $w_0 = w_1 = w_2 = w_3 = 0.25$. We obtain 20 realizations for each set of cross-sections. The main difference between the different settings is the improvement of computational efficiency with the increase in cross-sections. The proportion of pores (porosity) is reproduced at a similar level for each group, Also, when increasing the number of cross-sections $N_{cs}$, the number of geobodies gets closer to the reference, and the variability is decreased and the connectivity becomes stable, which are caused by the increase of conditioning data (i.e. informed cross-sections). On the other hand, using too many cross-sections will lead to a number of artifacts since the training sub-sections for each sub-block are very small, resulting in insufficient number of samples (see the sections extracted from the reconstructions in Figure 5). As a consequence, we recommend that several sections can be chosen if there are abundant candidates in one direction, which must ensure that the features of selected ones are diverse and contain enough spatial patterns, but not incurring artifacts. In this test, 3 or 4 sections in each direction are recommended, but it is related to the size of simulation grid in other 3-D application. In general, one informed section for every 50 grid cells in one direction in the simulation grid is recommended. When there are very few or no sections in a direction, a feasible solution has been suggested by *Gueting et al.* (2017) where sequential 2D simulations are performed to obtained some sections first, and then both the original informed data and the obtained sections are used to reconstruct the model of the entire 3-D domain.

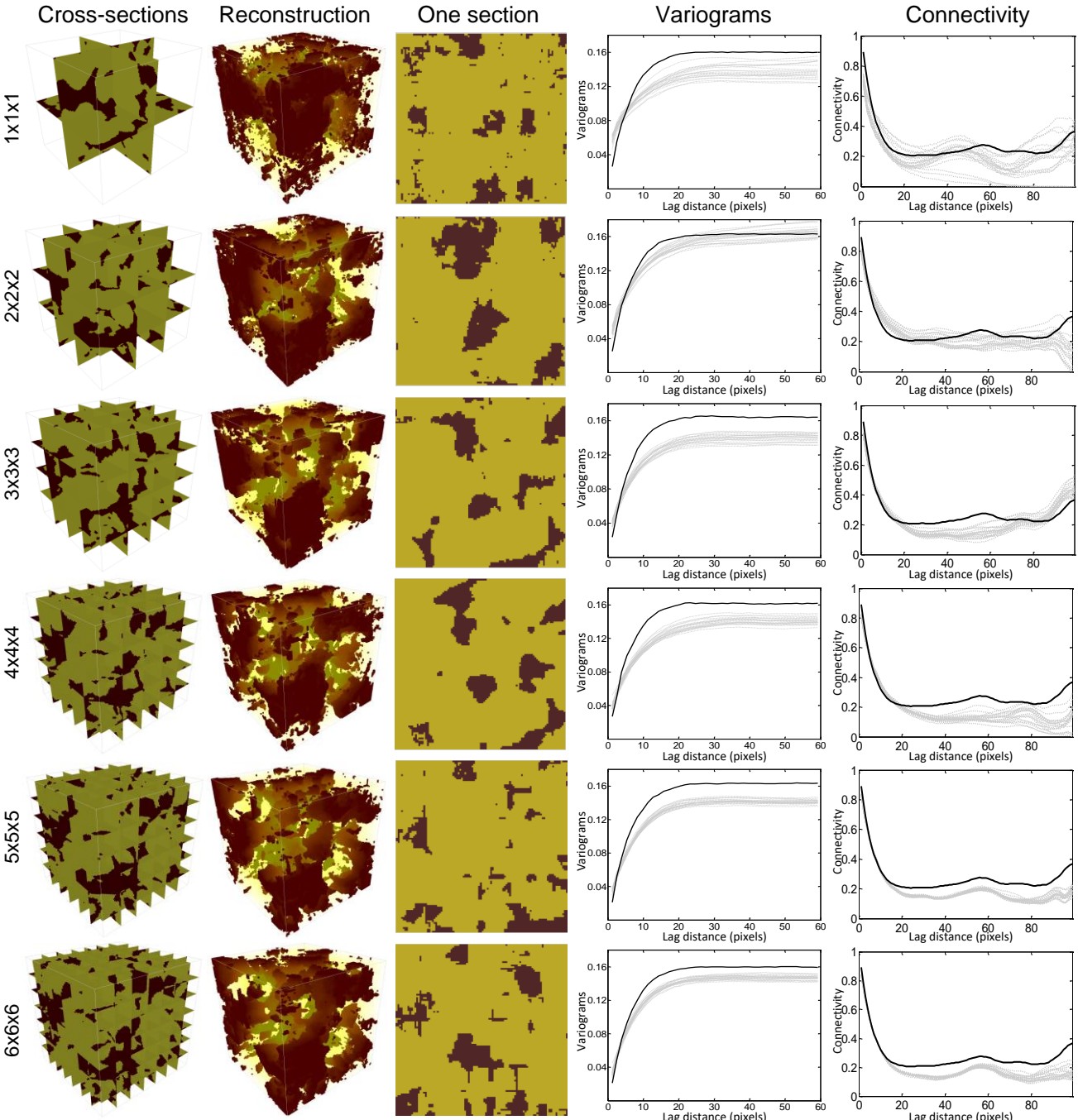

**Figure 5.** Reconstructions and their statistical properties with increasing the number of sections in each direction. The first section along X direction of a reconstruction for each case is presented here. We only present the connectivity functions computed along the coordinate Y since their features are similar in three directions. The black lines represent the corresponding features of the reference models, and the gray lines represent the features of the reconstructions.

**Table 1.** Comparison of the performance of the tests in Figure 5. All the statistics are the averages of 20 realizations.

| Test | $N_{cs}$ | Sub-blocks | Porosity (%) | | No. of geobodies | Time (s) |
| --- | --- | --- | --- | --- | --- | --- |
| | | | Training sections | Results | | |
| 1×1×1 | 3 | 8 | 19.07 | 16.36 | 1781 | 1382 |
| 2×2×2 | 6 | 27 | 21.35 | 19.95 | 908 | 718 |
| 3×3×3 | 9 | 64 | 18.70 | 16.22 | 572 | 396 |
| 4×4×4 | 12 | 125 | 19.62 | 16.21 | 471 | 271 |
| 5×5×5 | 15 | 216 | 19.81 | 16.80 | 340 | 183 |
| 6×6×6 | 18 | 343 | 19.74 | 17.32 | 326 | 127 |
| **3-D Ref.** | | | | **20.33** | **144** | |

### 4.1.2. Maximum Number of Matched Patterns from Each Training Image

Table 2 shows the statistics of 20 realizations obtained by varying the maximum of matched patterns from each training image $N_{max}$, which is a novel parameter adopted in this work to avoid the unnecessary searches during obtaining a cpdf from training images. Other parameters are the same as in the former test presented in Figure 5, except for the sections in each direction which are fixed to 3. We find that the computational cost increases sharply when $N_{max} > 160$ and then stabilizes. Concerning the compared statistical properties, low values of $N_{max}$ result in variabilities because it is almost like sampling the result directly from training images and the role of cpdfs is lost. For the remaining cases, the statistics are similar except for a decrease of variances with increasing $N_{max}$ (Table 2). In order to better grasp the effect of $N_{max}$, three cases are selected ($N_{max}$ = 5, 40, 320) and the corresponding realizations are shown in Figure 6a-c. The connectivity functions vary in a large range for small $N_{max}$ values. Conversely, they become more stable when increasing $N_{max}$ (Figure 6d). The variance of variables bears the same tendency by increasing $N_{max}$ (Figure 6e). Consequently, $N_{max}$ = 40 to 160 is recommended resulting in a balance between a stable cpdf and computational cost.

**Table 2.** Comparison of the performance for 20 realizations with three sections in each direction, and varying the maximum of matched patterns from each training image $N_{max}$. Other parameters are fixed and are same with the test of Figure 5. All the statistical values are the mean of 20 realizations. $\infty$ represents no constraint for $N_{max}$.

| $N_{max}$ | Porosity (%) | Variance | No. of geobodies | Time (s) |
| --- | --- | --- | --- | --- |
| 5 | 18.39 | 0.150 | 365 | 132 |
| 10 | 17.22 | 0.143 | 440 | 161 |

| | | | | |
|---|---|---|---|---|
| 20 | 16.69 | 0.139 | 486 | 200 |
| 40 | 16.47 | 0.138 | 505 | 251 |
| 80 | 16.48 | 0.138 | 510 | 417 |
| 160 | 16.38 | 0.137 | 519 | 495 |
| 320 | 16.50 | 0.138 | 503 | 549 |
| 640 | 16.66 | 0.139 | 508 | 587 |
| ∞ | 16.89 | 0.138 | 497 | 589 |
| **Ref.** | **18.70** | **0.152** | **144** | |

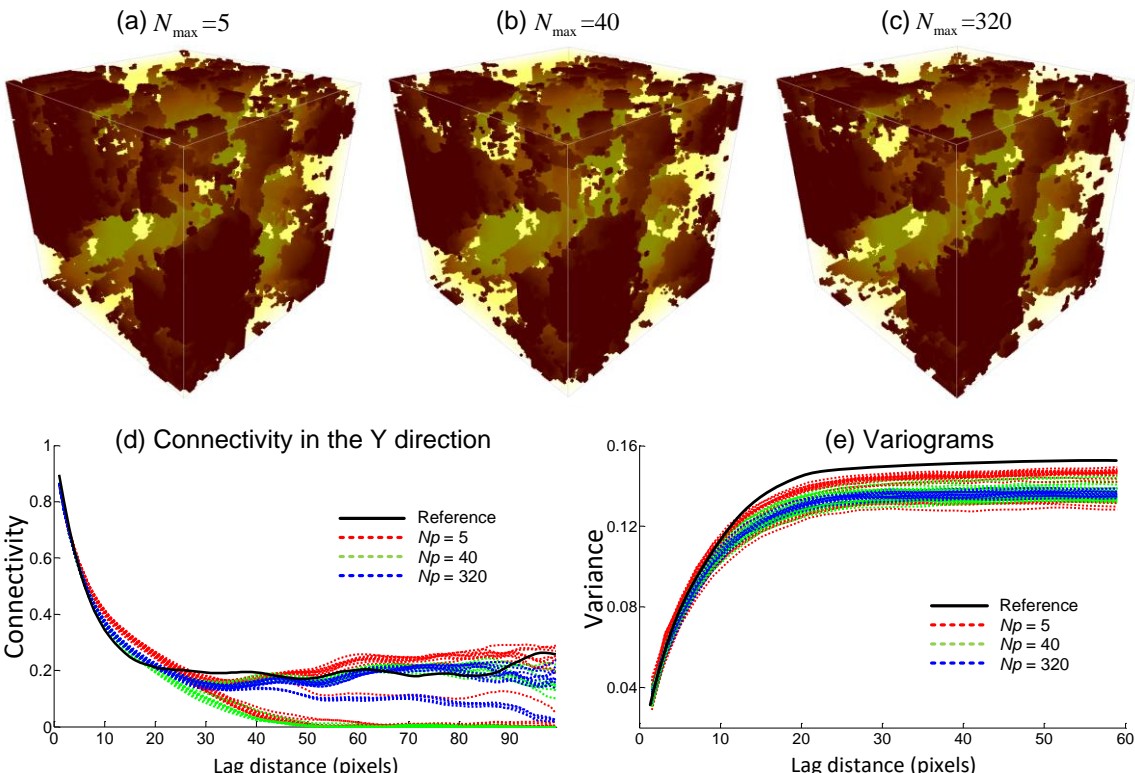

**Figure 6.** Reconstructions and their statistical properties with $N_{max}$ = 5, 40, 320 selected from Table 2.

### 5 4.1.3. Weights of the Probability Aggregation Formulas

In this work, the strategy for aggregating the pdfs from local sub-sections includes two steps. In the first step the weights of Linear Pooling Formula for two parallel sub-sections are selected depending on the distances between the current location and the two sub-sections in the first step. Therefore, the weights are automatically set and do not need to be set. In

the second step, the appropriate weights for the prior probability distribution and three orthogonal cpdfs are to be selected by the user. Figure 7 shows different realizations obtained by varying the four weights $w_0, w_1, w_2, w_3$. Here we increase the weight of the prior probability distribution $w_0$ and let the other three weights equal, since the cpdfs from three orthogonal directions have the same contribution. Of course, if users think the cpdf of one direction is more important than others, they

5  can be changed, under the constraint that $\sum_{i=0}^{3} w_i = 1$. It can be observed that when $w_0 = 0$, the spatial structures are well reproduced, but with larger variance (Figure 7a) since all spatial patterns are inferred from the MP statistics of the surrounding sub-sections rather than using prior information. When increasing $w_0$, the connectivity of the spatial structures is degraded, but the facies proportions are closer to the reference (Figure 7b). Finally, in the extreme case of (Figure 7c) the connectivity of spatial structures is lost. Therefore, $0 \leq w_0 \leq 0.25$ is a recommended range and other three weights can be

10  determined by the importance (e.g. complexity or variety of patterns) of the sections in each direction.

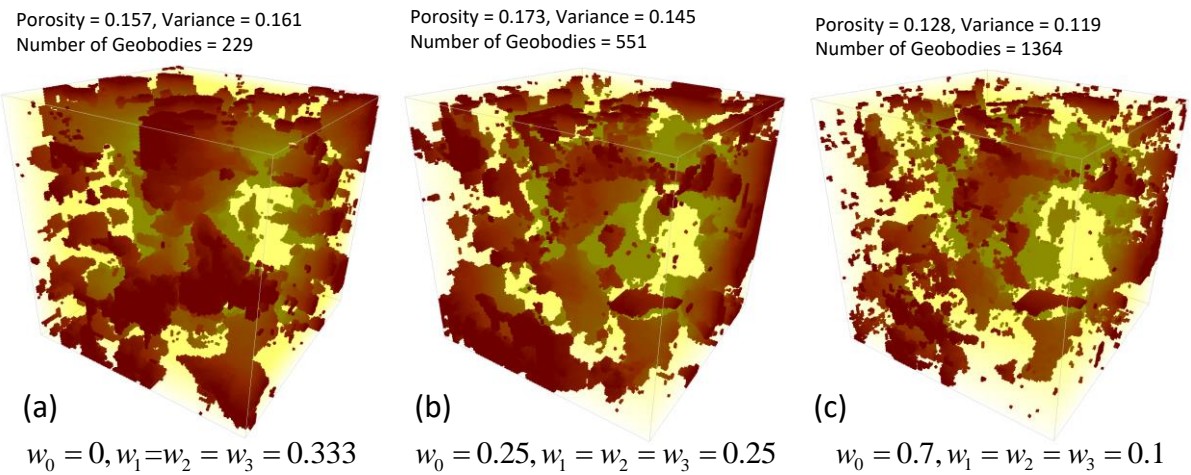

Porosity = 0.157, Variance = 0.161
Number of Geobodies = 229

Porosity = 0.173, Variance = 0.145
Number of Geobodies = 551

Porosity = 0.128, Variance = 0.119
Number of Geobodies = 1364

(a)  $w_0 = 0, w_1 = w_2 = w_3 = 0.333$

(b)  $w_0 = 0.25, w_1 = w_2 = w_3 = 0.25$

(c)  $w_0 = 0.7, w_1 = w_2 = w_3 = 0.1$

**Figure 7.** Three realizations obtained by varying the weights of the probability aggregation formulas. Three sections in each direction are used and other parameters are same with the test of Figure 5.

For the other parameters involved in our algorithm, most of them are similar to the parameterization of DS which have been tested thoroughly in *Meerschman et al*. (2013). However, 3DRCSs allows larger initial values for the neighborhood size and the distance threshold because multiple grids are used so that these initial values are decreased with increasing the level of multiple grids.

### 4.1.4. Interaction between $t$, $f$, $N_{cs}$ and $N_{max}$

In this section, we compare the interaction between two important parameters of DS (distance threshold $t$ and fraction of training image to scan $f$) and two new parameters presented in 3DRCSs (number of cross-sections $N_{cs}$ and maximum number of matched patterns from each training image $N_{max}$). Figure 8 shows the interaction between $t$, $f$, $N_{cs}$, $N_{max}$.

5 Running our algorithm with $f = 0.2$ and $t = 0.4$ results in noisy realizations. This is not surprising since any patterns can be accepted even if it bears a large pattern distance $d\{\mathbf{N}_X, \mathbf{N}_Y\}$. Of course, the algorithm will be very fast under these parameters because the scan for training image will be stopped at the beginning. *Meerschman et al.* (2013) tested thoroughly for the parameterization of DS. In their test, when $f = 0.5$ and $t = 0.2$, the realizations are acceptable. However, here the results still contain many noises since the local search strategy reduces the size of the actually scanned training images. As

10 the increase of $t$, $f$, $N_{cs}$ and $N_{max}$, the results become satisfactory. The recommended range of $N_{cs}$ and $N_{max}$ have been given in the above sections. In 3DRCSs, it is advised to use $f \geq 0.8$ and $t \leq 0.1$. Compared to DS, more strictly restrictions for $t$ and $f$ are adopted due to the local search strategy. Same as the effect of $t$ and $f$, $N_{cs}$ and $N_{max}$ also control the computational efficiency and the quality of simulations. Therefore, when setting the parameters, we should consider finding a balance between the quality of results and the computational cost.

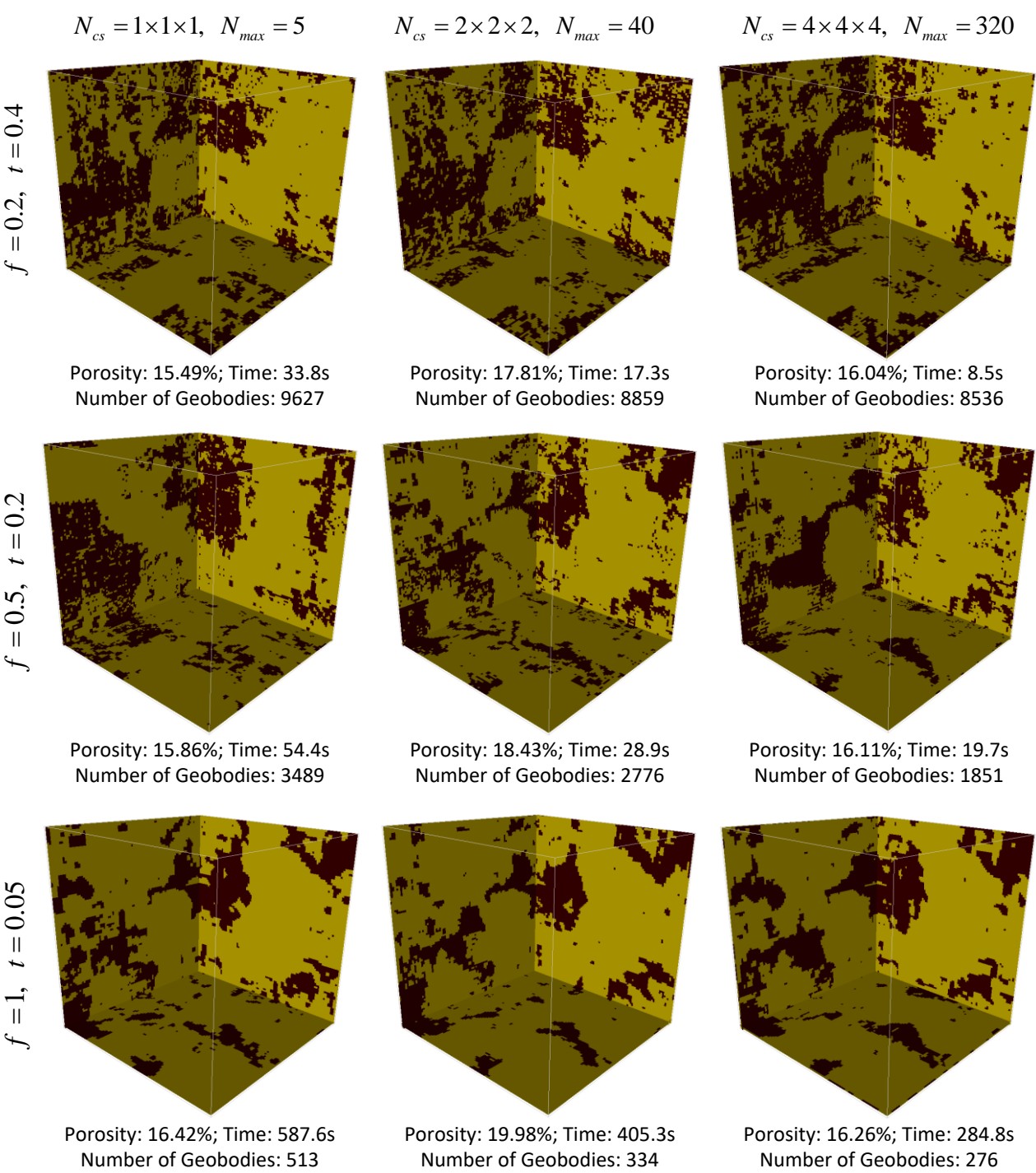

**Figure 8.** Interaction between $t$, $f$, $N_{cs}$ and $N_{max}$. These first sections in three directions of each realization are presented. The porosity, CPU time and the number of geobodies are the average of 10 realizations.

## 4.2. Comparison of Reproducing Heterogeneities with Existing Methods

To verify the validity of the approach 3DRCSs for reproducing heterogeneous structures, we compare it with two MPS implementations that use partial data: DS (*Mariethoz and Renard*, 2010) and s2Dcd (*Comunian et al.*, 2012). As shown in Figure 9, six cross-sections extracted from a 3-D model of folds (180 ×150 × 120 voxels) (*Mariethoz and Kelly*, 2011) are utilized in this test. s2Dcd is a wrapper library that requires an external MPS engine. In order to ensure comparability, here DS is employed as the engine of s2Dcd. The detailed parameters are as follows: maximum search radius = 40, maximum number of points in a neighborhood = 40, distance threshold $t_0 = 0.2$, fraction of training image to scan $f = 0.8$, maximum of matched patterns from each training image = 100, level of multiple grids = 3, weights of the probability aggregation $w_0 = w_1 = w_2 = w_3 = 0.25$. In other two methods, a smaller distance threshold $t = 0.05$ is considered and other essential parameters are same with 3DRCSs. Because the implementation of DS is parallel, we use 4 processors to carry out this test in DS and s2Dcd. Only one processor is used in 3DRCSs because our implementation is not parallel. In Figure 9, one selected realization for each method is presented. From their visual appearance, it looks that s2Dcd and 3DRCSs have the similar performance for reproducing the patterns shown in 3-D reference and informed cross-sections. Therefore, histograms, variograms, and connectivity functions are used to further analyze the performance. Figure 10 shows the comparison of proportions of the facies for the realizations by using three MPS methods. 20 realizations are performed for each method. It can be seen that the facies proportions with 3DRCSs are closer to the proportions of the reference model and the informed cross-sections. The variograms and the connectivity functions on three directions for the 3-D reference and the generated 20 realizations of each method are shown in Figures 11 and 12, indicating that all three methods are able to reproduce the basic statistics of the 3-D reference, but the lines of the proposed method are generally closer to the reference.

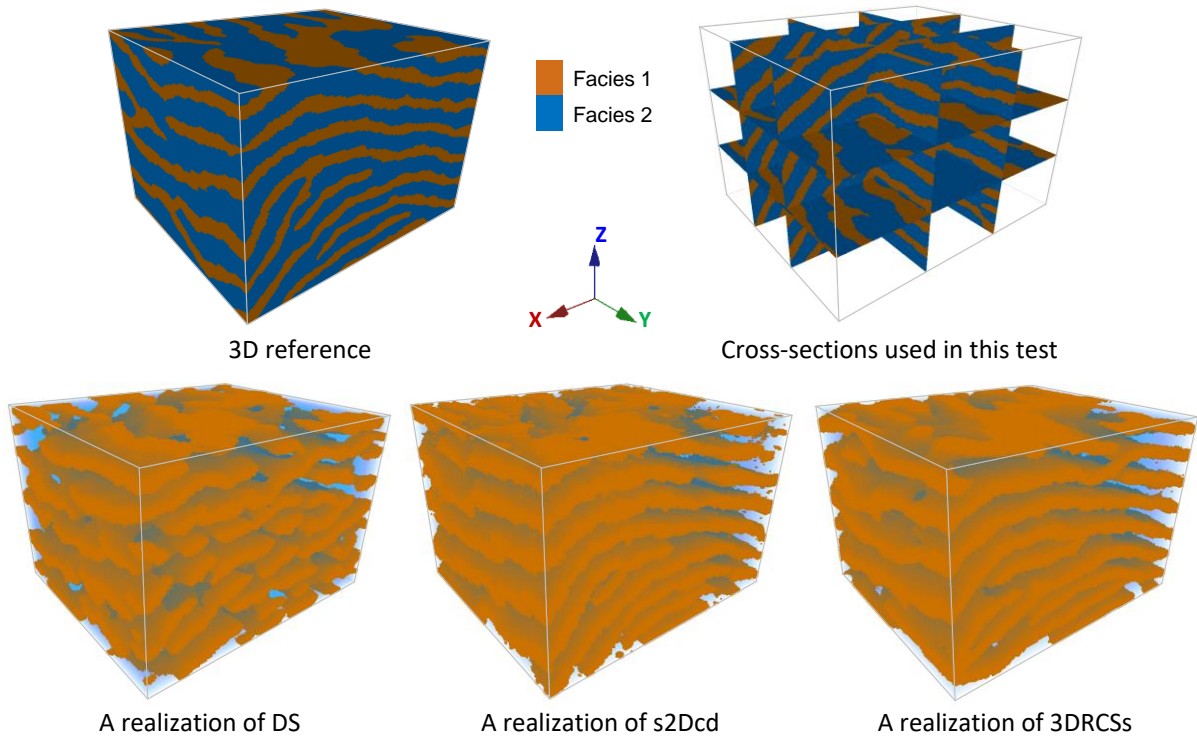

**Figure 9.** Realizations of three different MPS reconstruction methods.

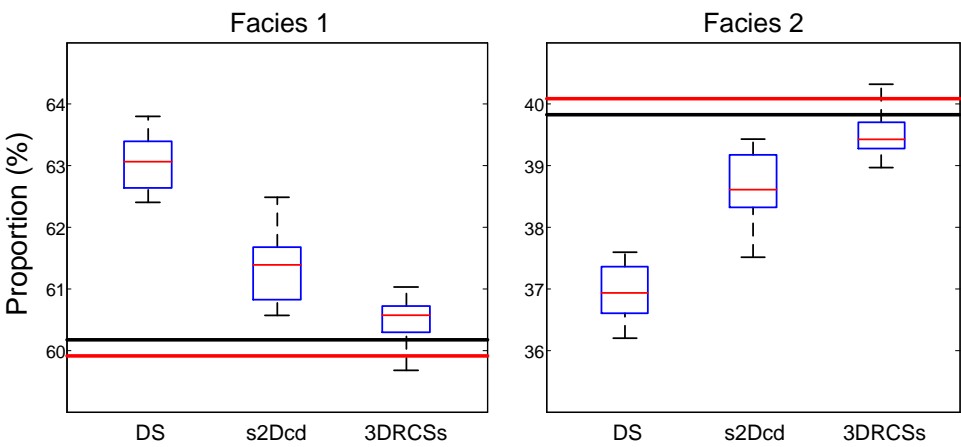

5   **Figure 10.** Proportions of the facies for 20 reconstructions by using three MPS methods. The black and red horizontal lines represent the proportions of facies in the 3-D reference and the cross-sections used as training images respectively.

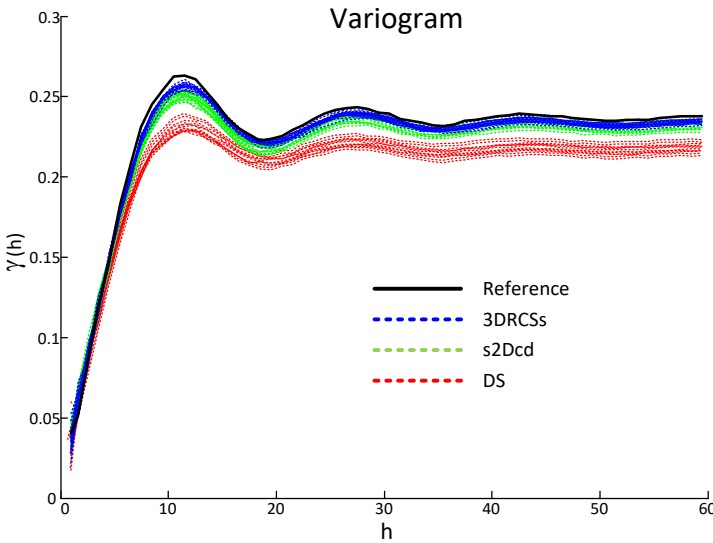

**Figure 11.** Comparison of the variograms between DS, s2Dcd and 3DRCSs.

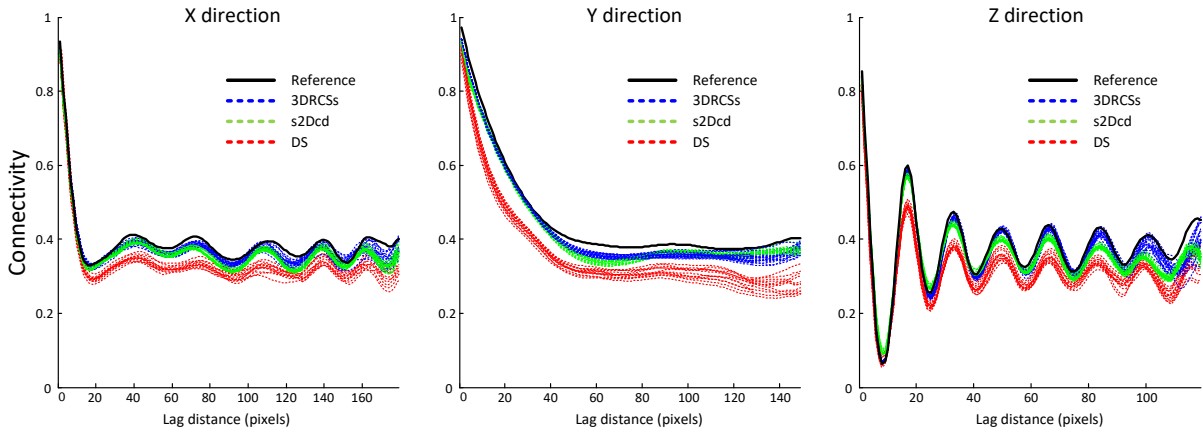

**Figure 12.** Comparison of the connectivity functions in three directions with three MPS methods.

To further compare the models obtained using the three different MPS approaches, MDS plots are constructed by calculating the distance of MP histograms between all the realizations of three approaches and a 3-D reference. The resulting MDS map is shown in Figure 13 and it can be observed that the realizations of 3DRCSs are closer to the reference in the MDS map than the results obtained by the other two approaches. In addition, kernel smoothing is used to estimate the density distribution of the realizations of three different MPS approaches around the reference. The probabilities of the realizations are calculated from kernel density estimation by using the equation (4) described in section 2.3. According to the reference model, the three different approaches have quite similar probabilities with 29, 33, and 38% for DS, s2Dcd, and 3DRCSs, respectively. However, the approach 3DRCSs still gains the highest probability.

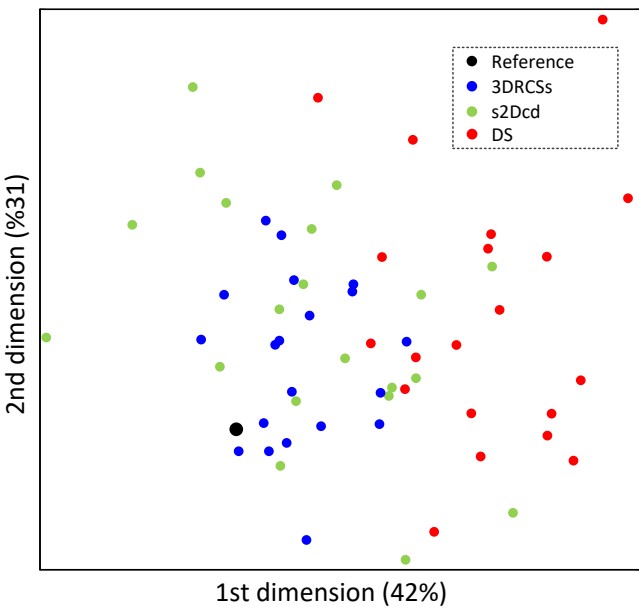

**Figure 13.** MDS representation for 20 realizations of each MPS method.

In practice, there is no fully informed 3-D reference and we only have several informed cross-sections. Thus, the statistical features of the reconstructions (e.g. variograms, connectivity functions and MDS plots) are close to the reference but no one can surpass it in the above test. However, these comparisons are still able to validate the reproduction of spatial patterns for the different MPS approaches.

### 4.3. Computational Performance

Section 4.1.1 and 4.1.2 have already analyzed the influence of the number of cross-sections $N_{cs}$ and the maximum of matched patterns from each training image $N_{max}$. Section 4.1.4 tested the interaction between $t$, $f$, $N_{cs}$ and $N_{max}$. The results indicated that the effect of $t$ and $f$ on the computational efficiency in 3DRCSs is the same as in DS. The computational performance of other parameters has been assessed clearly by *Meerschman et al.* (2013). The weights of the probability aggregation formulas do not affect CPU time.

A comparison of computational performance between DS, s2Dcd and 3DRCSs is presented in Figure 14. Because 3DRCSs is sensitive to the number of input cross-sections, we offer two and four sections in each direction respectively, and the computational efficiencies are shown in Figures 14a and 14b with increasing the total number of grid cells. Other parameters are the same as the test in section 3.2. Note that a different time axis is used for DS-based reconstruction because it uses much more CPU time than the other two methods, even though four processors are used for DS-based reconstruction. As shown in Figures 14a, the approach 3DRCSs presents better computational performance than DS-based 3-D

reconstruction since the MP statistics are captured from a smaller domain composed of several 2-D sections in s2Dcd and 3DRCSs. Because four processors are used in DS and s2Dcd, thus 3DRCSs presents the speedups of about 4 compared to s2Dcd and about 120 compared to DS in this test (Figures 14a). When increasing the number of cross-sections, the search space is divided into more subdomains in 3DRCSs so as to achieve a much better performance than s2Dcd and DS (see Figure 14b).

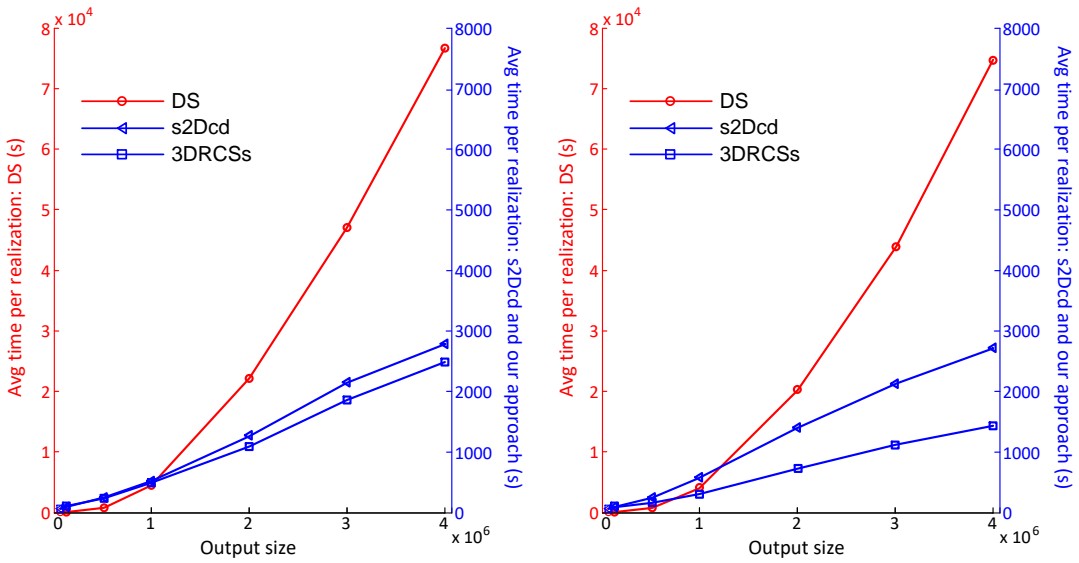

**Figure 14.** Comparison of computational performance between DS, s2Dcd and 3DRCSs with increasing the size of output grid: (left) two cross-sections and (right) four cross-sections in each direction. Note that different time axises are used in the two subplots, and four processors are used for DS-based reconstruction and s2Dcd but only one for 3DRCSs.

## 5. Synthetic Example: 3-D Reconstruction of Hydrofacies

To further demonstrate the applicability of our algorithm, an example from a real geological application is presented in this section. The Descalvado aquifer analog dataset (Figure 15) depicts the complex hydrofacies of a small area ($28m \times 7m \times 5.8m$) in Brazil (*Bayer et al.,* 2015). In the original dataset, there are five cross-sections derived from outcrops, which are marked by black lines in Figure 15a. They are referenced in a 3-D domain consisting of $280 \times 70 \times 58$ voxels. These sections allow creating only two parts of subdomains, which is insufficient for an application of 3DRCSs. Therefore, we borrow the strategy of *Gueting et al*. (2017) to insert three additional sections in yz direction using sequential 2-D MP simulation approach (s2Dcd) firstly which are marked by blue lines in Figure 15a. Then, all the tests are implemented on the basis of eight cross-sections (three in xz direction and five in yz direction) which are shown in Figure 15b.

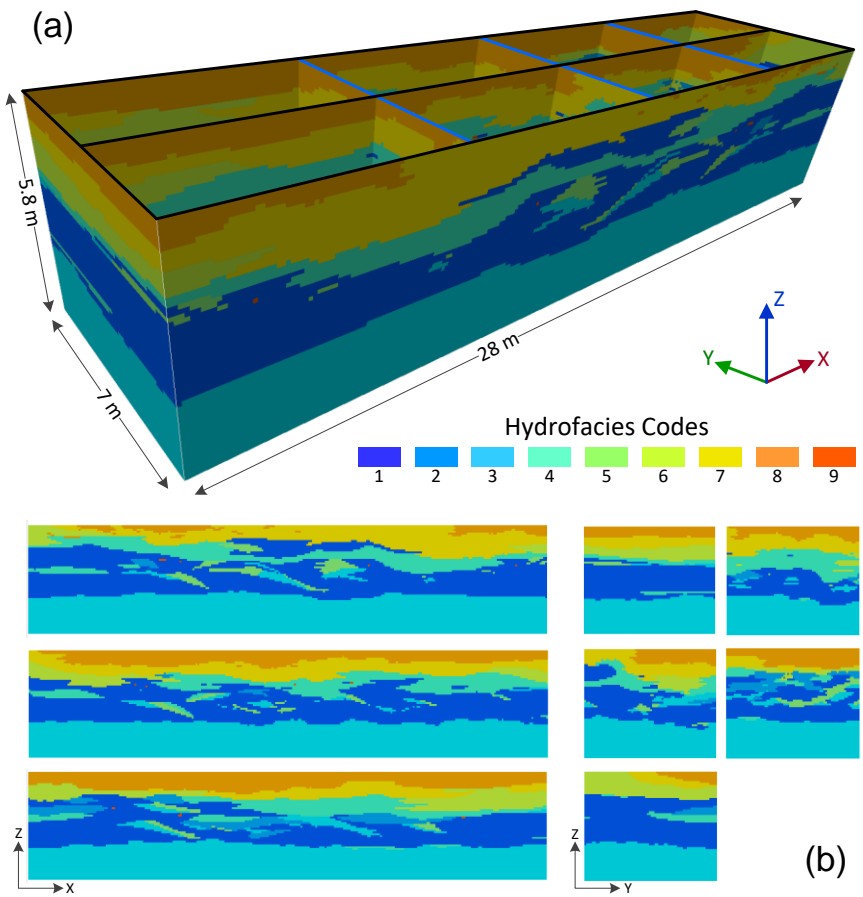

**Figure 15.** Descalvado aquifer analog dataset (*Bayer et al.*, 2015). (a) 3-D presentation of the informed cross-sections: three sections in xz direction, and five sections in yz direction; (b) 2-D presentation of these informed sections.

Figure 16 shows realizations obtained by using three different MPS approaches on the basis of the above-mentioned dataset. The white lines indicate the locations of informed sections in each realization. Note that an auxiliary variable along the *z* coordinate is used in s2Dcd and 3DRCSs. It is a continuous variable to control the changing trend of the hydrofacies along the *z* coordinate and the detailed description is given by *Comunian et al.* (2012). To further reveal the performance of the different approaches, we use MDS maps to visualize the dissimilarity of MP histograms (Figure 17), similarly as in

section 4.2. However here we use it to reveal the dissimilarity between all the reconstructed sections exacted from the realizations and the eight informed sections along the two directions, rather than different 3-D realizations. Thus, for each realization, 70 sections (67 reconstructed sections and 3 informed sections) from *xz* direction and 280 sections (275 reconstructed sections and 5 informed sections) from *yz* direction are used to draw the MDS maps along the two directions respectively. MDS is very appropriate to present the dissimilarity for this kind of applications because we only have partial

cross-sections instead of an entire 3-D training image. Therefore, it is necessary to assess the dissimilarity between the

reconstructed sections and informed sections. As shown in Figure 15, the sections from *xz* and *yz* directions are very different, such as the correlation lengths and the complexity of structures. Thus, we draw different MDS maps respectively for the *xz* and *yz* directions (Figures 17a and 17b). Individual sections from the realizations are compared in Figures 17c and 17d. Overall, it can be observed both visually and in the MDS maps that the sections obtained from 3DRCSs are closest to the informed sections.

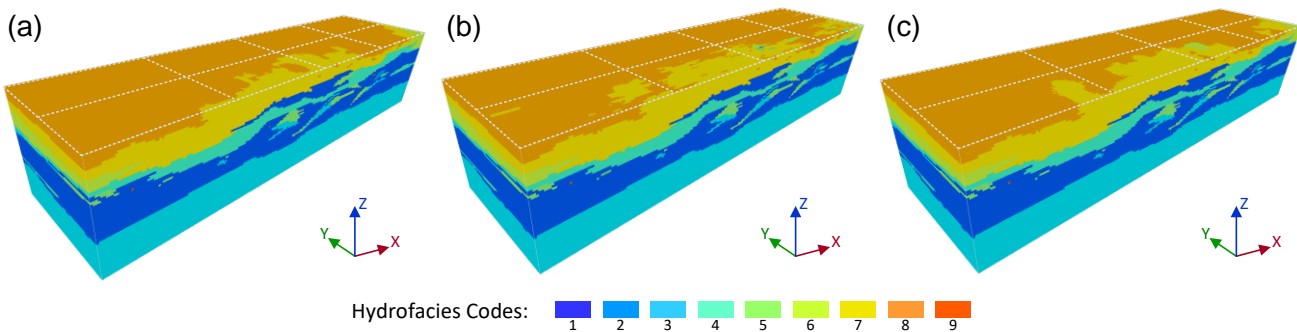

**Figure 16.** Three realizations using three different MPS approaches: (a) DS, (b) s2Dcd with the coordinate z as auxiliary variable; and (c) 3DRCSs with the coordinate z as auxiliary variable.

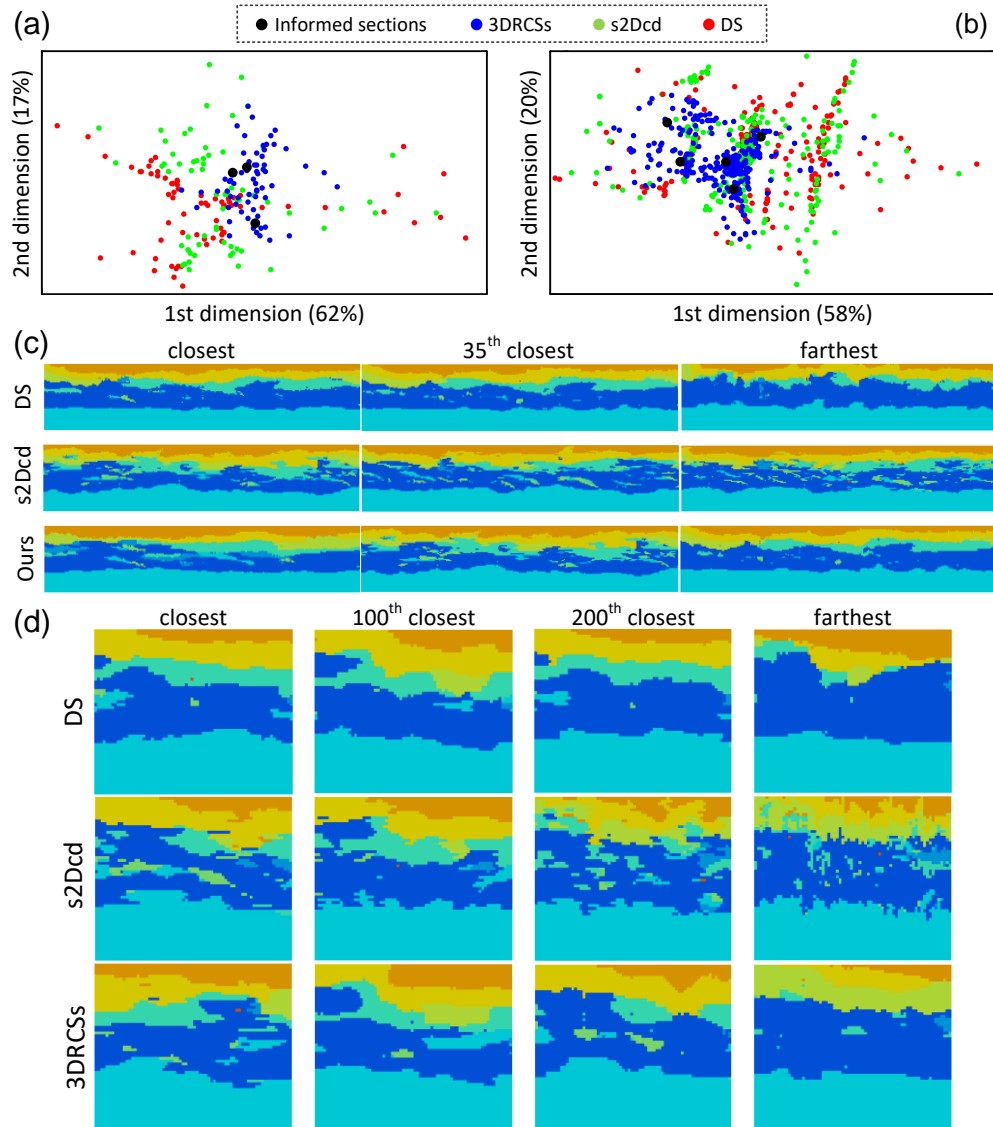

**Figure 17.** MDS maps of sections extracted from realizations using three different MPS approach. (a) MDS map of 70 sections for each realization along *xz* direction; (b) MDS map of 280 sections for each realization along *yz* direction; (c) selected sections for each method according to the JS divergence in *xz* direction; (d) selected sections in *yz* direction.

3DRCSs is able to reduce the non-stationarity effect of real geological data to a certain extent due to the local search strategy. As shown in the above analysis, the patterns in the informed cross-sections are very complicated where the distribution of hydrofacies is anisotropic and non-stationary, especially for the facies with a lower proportion. As illustrated in Figure 18a, a local domain is surrounded by four segments from the informed cross-sections. It should be noted that there

10   is no facies 2 in all the four segments. We extract the local parts from three realizations by using different MPS approaches.

Then we check all the segments of the three local models, and we find that facies 2 is reproduced in this local area in the realizations of DS and s2Dcd. Three segments are randomly selected from the three local models, and they are shown in Figure 18b where the boundaries of facies 2 are marked by red lines. Figure 18c shows the histograms of the four informed segments and the local models of 10 realizations for each MPS method. It can be observed that, although there is no facies 2 in the closest four segments, it is reproduced in this local area by DS and s2Dcd. Conversely, 3DRCSs can maintain the distribution of facies well since all the MP statistics are captured from the surrounded sub-sections. If the surrounding sub-sections of a local area do not contain an attribute but it exists in other locations, the patterns with this attribute will not be moved to this local area in the approach 3DRCSs. This indicates that 3DRCSs allows involving the non-stationary geological analogs in the 3-D real applications, and spatial patterns are restricted into a local domain so that they are not carried to faraway locations.

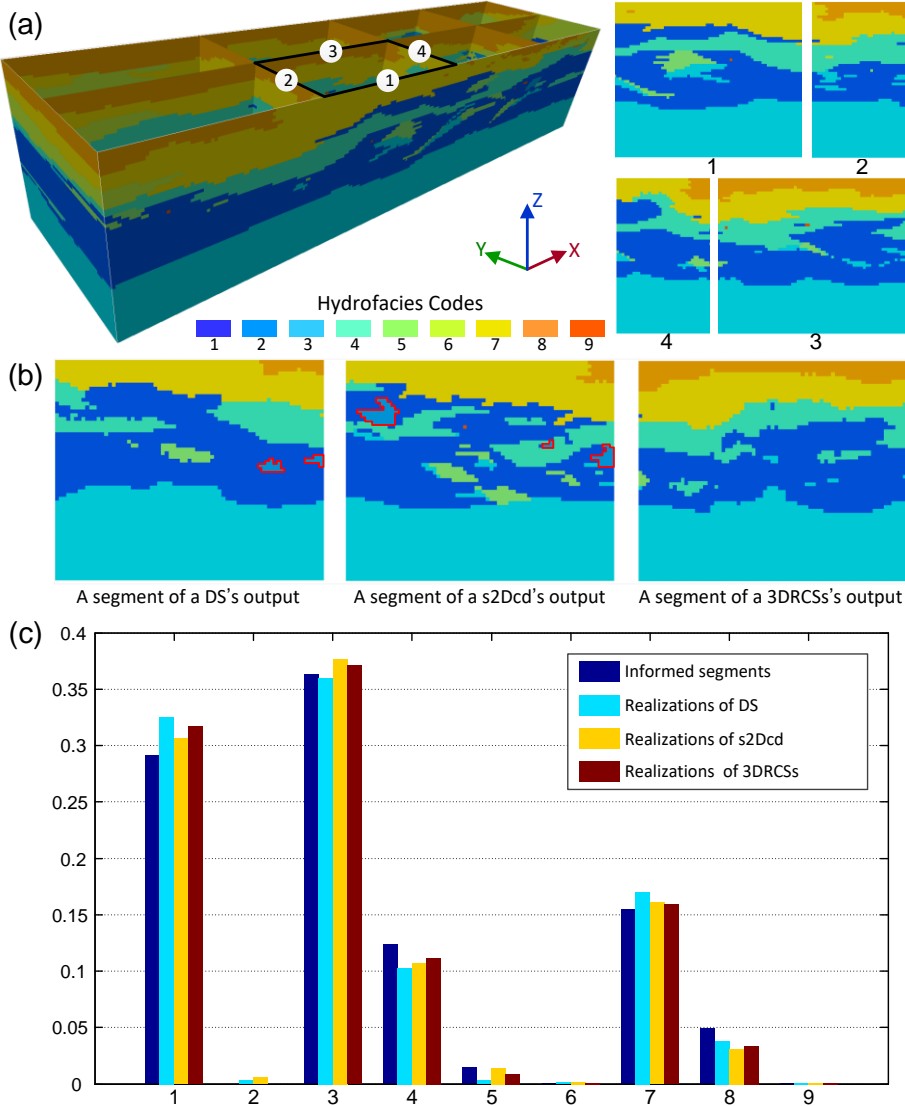

**Figure 18.** Comparison of reproduction of non-stationary patterns. (a) a local domain and the four corresponding segments; (b) three selected segments from the realizations obtained using different approaches in the local area; (c) histograms of the four informed segments and the local models of 10 realizations for each MPS method.

In the real-world applications, the geological sections or other analogs are not always straight or orthogonal. Therefore we need to project them in orthogonal directions. Figure 19 illustrates the process of projecting the tortuous sections to the parallel planes along a given direction. The same strategy can be used to address the issues in other directions. After that the original sections will be used as hard data and the projected sections will be only as training images. Thus other scattered samples (e.g. boreholes, outcrops) also can be involved as hard data.

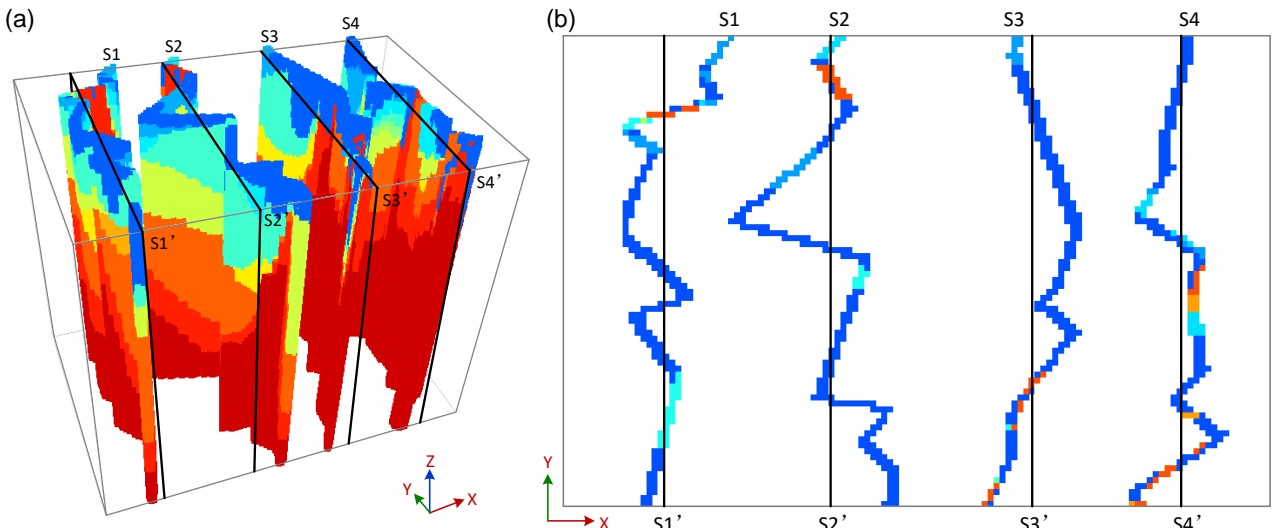

**Figure 19.** Process of projecting real-world sections to parallel planes along a given direction: the process in (a) 3-D space and (b) xy plane.

## 6. Discussion and Conclusion

In this paper, we presented a novel method (3DRCSs) for reconstructing 3-D complex heterogeneous structures by using partial lower dimensional data. Indeed, this is a very general issue since inferring high-dimensional patterns from low-dimensional data (e.g. boreholes, outcrops and other analogs) is a very common workflow for geologists. In practice, reliable 3-D models of complex geological structures are still difficult to construct due to the heterogeneity of geological phenomena and processes, even though there are many real geological analogs or sections that can be used. 3DRCSs makes it possible to reconstruct 3-D structures with MPS when no 3-D training image is available. The synthetic experiments and practical applications presented in this paper demonstrate the capacity to reconstruct such heterogeneous structures.

As compared to the previous MPS implementations that use partial data, the proposed method requires several local training sub-sections surrounding a simulated node, rather than a full section (*Comunian et al.*, 2012) or points in a 3-D domain (*Mariethoz and Renard*, 2010). The local search strategy proposed in this paper allows to compute more reliable MP statistics because it avoids that spatial patterns from faraway locations are considered in the simulation of the current node. In this strategy, the original cross-sections are divided into many sub-sections according to their spatial relationships. Therefore, the non-stationarity of real geological analogs is reduced to a certain extent because the training patterns cannot be borrowed from further than a local subdomain. Of course, besides cross-sections, other scattered samples also can be included as hard data.

Moreover, 3DRCSs increases the computational efficiency compared with existing MPS methods. The local search strategy allows acquiring MP statistics from the local sub-sections so that the searches are significantly reduced. Its good

computational performance makes it potentially applicable to real 3-D modeling problems such as porous media, hydrofacies, reservoir, and other complex sedimentary structures. In addition, a new parameter, the maximum of matched patterns from each training image is adopted to avoid the unnecessary searches. The experimental results demonstrated that a reasonable choice for this parameter can not only ensure to capture a stable cpdf, but also gain a further performance speed-up.

The method presented here retains many advantages of DS (*Mariethoz et al*., 2010), such as unnecessary storing for MP statistics, pattern distances, flexible neighborhood. Nevertheless, we propose an adaptive and flexible implementation of the search template on multiple grids where the radius of the neighborhood, the distance threshold and the size of data events decrease linearly with the rising of levels of multiple grids. As a result, a big data event is divided into several small parts placed on the different grids, which results in a smaller neighborhood on each grid. An acceptable distance threshold is

assigned to the first grid to make it easier to obtain a stable cpdf and to capture the large-scale features from the original sparse samples. For the last grid, the radius of neighborhood is reduced to one and the highest criterion is carried out for the threshold (i.e. $t = 0$) which avoids the small-scale features or lower proportion facies are filtered out. Hence, the simulation of each multi-grid is simulated with different parameters, allowing for flexibility in simulating different structures at different scales.

Another important advantage of 3DRCSs is the probability aggregation strategy where the combinations of two different formulas are used to combine the cpdfs from different sub-sections. First, an additive aggregation method, linear pooling formula is used to combine two disjunctive probability distributions from each pair of parallel sub-sections to obtain a more stable pdf. The weights of this step are related to the distances between the current location and the two parallel sub-sections. Such parameterization is able to ensure the pattern trend changing from one sub-section to another one. And then,

we aggregate the orthogonal pdfs and prior probability distribution by using a multiplicative method, log-linear pooling formula. This step can enhance the capability for reconstructing connectivity of spatial patterns in comparison with the method using a series of 2-D MPS simulations to fill a 3-D domain along given orthogonal directions (*Comunian et al.*, 2012).

      The limitations of the method 3DRCSs come from that it is not always possible to obtain abundant sections in each

direction, and extremely small local blocks cannot offer enough spatial patterns, thus a minimal sub-section size has to be considered. In addition, 3DRCSs is not able to perform the simulation of continuous variables. The proposed method can be further improved to overcome these limitations. Another possible direction is to parallelize the proposed MPS implementation and further enhance its computational performance.

*Competing interests*. The authors declare that they have no conflict of interest.

## Acknowledgments

We are grateful to Thomas Hermans, Kashif Mahmud and one anonymous referee for their insightful comments and suggestions towards improving the research enclosed in this paper. This work was supported in part by the National Natural Science Foundation of China (U1711267, 41172300) and the Ministry of Education Key Laboratory of Geological Survey and Evaluation (CUG2019ZR03). The authors wish to thank Philippe Renard and Julien Straubhaar for providing the MPS algorithm DeeSse; Moctar Dembele, Min Zeng and Luiz Gustavo Rasera for the fruitful discussions. An executable program of the proposed algorithm is available on the website of the first author (http://www.escience.cn/people/chenqiyu/index.html), and the source code developed by using C++ is available on request from the first author (qiyu.chen@cug.edu.cn).

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
