# Peer review of "Locality-based 3-D multiple-point statistics reconstruction using 2-D geological cross-sections"

_Hydrology and Earth System Sciences, 2018_

## Referee Comment (RC1) · Anonymous Referee #1 · 24 Jun 2018

This paper presents another approach to handle the problem of lacking three-dimensional training images for multiple-point geostatistical simulations. The authors explain the differences between their approach and previously existing ones and perform thorough sensitivity analyses on the new parameters required by their implementation. However, they take some prior important decisions that go unchallenged, such as the choice of the probability aggregation methods, and the parameters used in them. A sensitivity analysis of these decisions would provide a more solid basis for the usage of the new approach.

My major criticism is on the tests performed during the benchmarking. They are purely statistical, yet this is a journal very much related to surface and subsurface hydrology. Readers of HESS and potential users of this method would be more appealed to use it

if the benchmark would include, for instance, some solute transport simulations.

---

## Referee Comment (RC2) · Anonymous Referee #2 · 31 Jul 2018

Dear Chen et al.,

I read with interest the paper entitled "Locality-based 3D multiple-point statistics reconstruction using 2D geological cross-sections". The paper describes a new methodology to build 3D realizations constrained by 2D sections that act as local training images and hard data for MPS algorithms. The method seems to give realistic results and perform better than existing algorithms. Globally, the paper and the presented results are convincing. However, I would like the following general comments to be addressed by the author:

1) The introduction is a little bit messy, giving an overview of many papers related to MPS, but missing the point of highlighting the specific issues tackled by the paper. For example, the 2nd paragraph (P2L15) makes the history of MPS. This is not the point,

you should rather insist on the importance of the TI (what you do in paragraph 3) and the description of the state of the art (relatively vague in the current version). Many sentences in the introduction are too vague such as "some assumptions have been implemented to reconstruct 3D models" (P3L14, which assumptions?) or "A promising reconstruction method . . . by adapting the DS algorithm. However, large-scale" (P3L16, the "however" refers to something that is not explained, the problem should be clearly stated).

2) I found the methodology section difficult to follow. Indeed, the proposed approach borrows some techniques from existing algorithms (mostly direct sampling), so that part of the methodology is described in other papers. Although, some parameters are common with DS, the philosophy is quite different as DS is never explicitly calculating cpdf. In that sense, the proposed methodology is closer to classical approaches such as snesim (except that the cpdf are not stored). I would therefore recommend to explicitly describe every part of the algorithm, without (too many) references to previous publications. Doing so, the methodology would be self-sufficient. Part of the methodology (dissimilarity metrics and MDS) is introduced in the result section and should be moved to the methodology.

3) In the parameter sensitivity, it is argued that some parameters are similar to DS, and thus the paper focuses on the new parameters. Nevertheless, although the proposed method borrow some ideas from DS, it is clear that the approach is different (In DS, you stop searching as soon as you find an occurrence with distance below the threshold, here you continue scanning to get the cpdf). Therefore, we cannot assume without checking that some parameters such as the threshold, the search neighborhood or the fraction of the TI that is investigated will have the same influence in DS and in the proposed approach. In addition, some interaction between the parameters is expected. For example, between the threshold and the maximum number of occurrence, or the number of available sections, some interactions can be expected. Indeed, if you increase the threshold, you will reach faster the maximum number of occurrence.

4) The application example is not a real application, but another synthetic benchmark using a real analog. Indeed, there is no verification data nor specific application of the model. I would therefore merge section 3 and 4.

Specific comments:

1) P3L6-25. You don't discuss the paper by Gueting et al. (2017) in the introduction, although it is a recent paper on the topic and you borrow some ideas later in the field example. You should reconsider this paragraph to describe with more details previous approaches and how your method is new.

2) P4L29-30. This sentence does not necessarily refer to the approach proposed by Comunian et al. (2012), but more globally, to any method assuming stationarity. It is always possible to use auxiliary variable to account for non-stationarity, as it is done by the s2Dcd approach in the application example.

3) P5L3-8. For reconstruction algorithm, there might be a confusion between training image and hard data. Here, the sections that are used are both training images and hard data. To some extent, using the whole section as TI and the sub-sections as hard data is already a good way to locally constrain the simulations. The S2Dcd algorithm is already performing well in that sense.

4) P5-L9-10. The choice of the subsections within domains is not necessarily selecting the most local information. Indeed, this depends on the location of the node within the subgrid, some other sections might be closer, for example when close to a domain boarder. Since the approach does not store any cpdf, you could center the sub-domain on the node and the TI would change for each node. This would avoid the issue of case 2 in Figure 2, where many nodes closer to the node to simulate are actually out of the TI.

5) P7L18-20. I am not sure it is intuitive, you might expect that the parallel sections are somehow correlated (except in case of strong non-stationarity), the multiplication of

probabilities could then make more sense... and otherwise for perpendicular directions (if the field is highly anisotropic, the orthogonal directions are bringing totally different information)...

6) P8L15-18. In short, you compute the cpdf using all the neighborhood whose distances are below the threshold, right?

7) P9L1-3. Multi-grid approach. The description is not clear. How do you divide the data event within several grids? I thought you were selecting all the data and previously simulated nodes within the radius. Are you only considering neighboring nodes that are on the current grid? Or you just mean that for the first grid there are less points simulated? In practice, everything depends on the radius. Please clarify.

8) P9L4-5. I understand that you take the "diagonal nodes" (figure 3) for the smaller grid as they are previously simulated in the multi-grid 2, but the other nodes should also be consider (horizontally and vertically) according to your radius, thus 8 nodes in 2D and 26 nodes in 3D, no? The remark about 3D is strange since you are only considering 3 directions in 2D, this is not a 3D neighborhood or am I missing something? How do you consider previously simulated points or hard data that are out of the 2D planes? Are they simply disregarded, or somehow projected on the plane?

9) P18. Figure 9. The red line shows the proportion in the cross-section, what about the "true" proportion?

10) P19L6-14. You could actually quantify how realizations are realistic using kernel smoothing (to average the information of the different realizations in a metrics), estimating the density distribution of the realizations around the reference. You can refer to the already cited paper of Hermans et al. (2015) for an application of MDS in the context of 3D MPS in hydrology.

11) P20L15-16. Still, in DS no cpdf is computed as the first matching sample is selected. Here, you continue scanning, so the effect of those parameter is not necessarily similar. For example, an interaction between t, the number of sections and Nmax is expected.

12) P22L6. You say that the number of sections is insufficient, but you don't provide guidelines about a sufficient density to use your algorithm. What if the sections are not oriented in orthogonal directions?

13) P23L2-3. A short description of the auxiliary variable would be welcome. I guess it describes the proportion of facies in the different zones along the vertical direction. It does not have to be long.

14) P26L2-3. "partial lower dimensional data" is not true. You can only use 2D orthogonal sections in a sufficient amount. Borehole or analogs cannot be used since you argue that local information is important.

15) P26L7. But you need a lot of 2D sections, which is a clear drawback of the method.

Technical comments:

1) P12L6-7. Does the "maximum search distance" correspond to the Radius of previous sections?

2) P21L9. You previously mentioned (P17L15) that S2Dcd was using DS with 4 processors. Please check.

3) P23L11. I think it should be Figure 17 instead of 16.

---

## Referee Comment (RC3) · Anonymous Referee #3 · 25 Aug 2018

This paper presents a locality-based MPS approach to reconstruct 3D geological models based on easily available 2D training images. To fulfil the objective, the MPS search engine roams over only several local sub-sections closer to the simulated node, instead of using a full training image. The authors also perform a parameter sensitivity analysis and the performance comparison with other previous 3D reconstruction techniques, illustrating the effectiveness of their approach using synthetic and real geological data. The results identify better performance both in portraying complex heterogenous structures and in CPU cost.

All together a very good paper, well written and showing a clear and valuable contribution that deserves publication. However, a number of significant issues need to be addressed for this manuscript to be publishable. Therefore, the authors are neverthe-

less invited to consider carefully the following comments to improve their manuscript.

General comments:

1. I am not totally convinced with the overall contribution of this method compared to s2Dcd. This needs to be explained in detail how the proposed technique differs from s2Dcd, which is now lacking in the introductory part.

2. The MDS shows slight improvement in terms of MP simulations using the proposed scheme. The computational benefit only appears with abundant sections available in each direction, which is in practice seldom existing and also mentioned as a limitation in the manuscript. Moreover, the improvement with reproduction of non-stationary patterns might have sampling effect as only one realization is considered from each method.

3. Overall, I am struggled to understand the flow of the methodology section, e.g. how the multigrid concept is implemented in searching the neighborhoods, or am I missing something in the workflow of the algorithm? I would also like to see the effects of using various number of multigrid in the form of sensitivity analysis.

Specific Comments:

1. P7 L2-3: Rewrite the sentence.

2. P12 L12: the connectivity 'becomes'

3. P12 L14: I would prefer to see an example of artifacts clearly visible on a section of the reconstructed model (maybe with the example of 6x6x6 model in Figure 5), to have the feeling of how bad it is and also to justify the logic behind not using too many cross-sections.

4. P12 L18: it 'is' related

5. Figure 5: Describe the black and gray lines by adding legend or in figure caption. I think the black lines represent the reference model? Also add the axes labels in

variogram and connectivity plots.

6. P14 L8-10: Rewrite the sentence as it's hard to follow in this format.

7. P14 L14: 120? or 160 or 320?

8. P 17 L19: analyze 'the' performance.

9. P 17 L21: our method

10. Figure 8: Caption is incomplete

11. Figure 9: The proportions of the facies in the 3D reference could be added as well in the plot for comparison.

12. P 20 L15-16: A brief summary of all other optimized parameters would be helpful for the readers.

13. Figure 13: The figure is redundant as all these numbers are already in the tables.

14. P 21 L9: s2Dcd uses DS as an external MPS engine as mentioned in P17 L15-16, therefore s2Dcd also runs on 4 processors, I believe. However, the authors claimed the opposite here. Please clarify.

15. P 22 L6: parts 'of' subdomains

16. P 23 L5-6: Figure 17 compares the dissimilarity between the sections extracted from the realizations and the informed sections, and I am guessing the sections are selected as random and the authors avoid the sections those are already used as training images?

17. P 23 L11: Figure 17 instead of Figure 16.

18. P 25 L1: The segments in Figure 17b are chosen from three local models, so is there any sampling effect when you select the sections to compare the reproduction of non-stationary patterns? What if you take an ensemble of sections from few realizations to compare the techniques?

19. P 25 L11: 'extracted'

---

## Author Comment (AC1) · 23 Sep 2018

The comment was uploaded in the form of a supplement:
https://www.hydrol-earth-syst-sci-discuss.net/hess-2018-256/hess-2018-256-AC1-supplement.zip

---

## Author Comment (AC2) · 23 Sep 2018

The comment was uploaded in the form of a supplement:
https://www.hydrol-earth-syst-sci-discuss.net/hess-2018-256/hess-2018-256-AC2-supplement.zip
* * *

---

## Author Comment (AC3) · 23 Sep 2018

The comment was uploaded in the form of a supplement:
https://www.hydrol-earth-syst-sci-discuss.net/hess-2018-256/hess-2018-256-AC3-supplement.zip

---

## Author Response (AR1)

**Response to Reviewer #1:**

This paper presents another approach to handle the problem of lacking three dimensional training images for multiple-point geostatistical simulations. The authors explain the differences between their approach and previously existing ones and perform thorough sensitivity analyses on the new parameters required by their implementation. However, they take some prior important decisions that go unchallenged, such as the choice of the probability aggregation methods, and the parameters used in them. A sensitivity analysis of these decisions would provide a more solid basis for the usage of the new approach.

We are grateful for your insightful and constructive comments and suggestions. In order to more clearly present the probability aggregation strategy proposed in our paper, we moved the description of two existing formulas used in our work to a new section Background Information in the revised version (see P5L18-26 and P6L1-11). Thus the section 3.2 mainly focuses on the strategy for aggregating the pdfs from local sub-sections proposed in this work (see P9L7-22 and P10L1-13). The choice of the probability aggregation methods is described in P9L13-20. The sensitivity analysis of the weights of the probability aggregation formulas has been done in section 4.1.3 (see P18L2-16).

My major criticism is on the tests performed during the benchmarking. They are purely statistical, yet this is a journal very much related to surface and subsurface hydrology. Readers of HESS and potential users of this method would be more appealed to use it if the benchmark would include, for instance, some solute transport simulations.

Although this manuscript mainly focuses on the algorithm principle of reconstructing 3-D models of subsurface structures by using multiple-point geostatistical techniques, several data sets in hydrology or hydrogeology are used to test our methods, such as pore structure of sandstone (see section 4.1), folds in subsurface aquifers (see section 4.2), and a synthetic example: 3-D reconstruction of hydrofacies (see section 5). Therefore, we think that our method can be used to reconstruct 3-D models of complex heterogeneous structures in hydrology or hydrogeology and it meets the scope of HESS.

**Response to Reviewer #2:**

I read with interest the paper entitled "Locality-based 3D multiple-point statistics reconstruction using 2D geological cross-sections". The paper describes a new methodology to build 3D realizations constrained by 2D sections that act as local training images and hard data for MPS algorithms. The method seems to give realistic results and perform better than existing algorithms. Globally, the paper and the presented results are convincing. However, I would like the following general comments to be addressed by the author:

Thank you very much for your positive and constructive comments and suggestions. We have corrected all the issues you raised in the revised version. The following is a point-by-point response according to your comments.

1.  The introduction is a little bit messy, giving an overview of many papers related to MPS, but missing the point of highlighting the specific issues tackled by the paper. For example, the 2nd paragraph (P2L15) makes the history of MPS. This is not the point, you should rather insist on the importance of the TI (what you do in paragraph 3) and the description of the state of the art (relatively vague in the current version). Many sentences in the introduction are too vague such as "some assumptions have been implemented to reconstruct 3D models" (P3L14, which assumptions?) or "A promising reconstruction method … by adapting the DS algorithm. However, large-scale" (P3L16, the "however" refers to something that is not explained, the problem should be clearly stated).

    Thank you very much for your constructive suggestions. We rewrote most of the introduction to highlight the specific issues tackled by our paper. Specifically, the history of MPS is compressed in the revised manuscript (see P2L4-L13, L22; P3L4-8); vague descriptions have been corrected (see P3L16-22); and the limitations of the existing have been depicted clearly in the new version (see P2L26-32, P3L25-26, P3L29-34, and P4L1-2).

2.  I found the methodology section difficult to follow. Indeed, the proposed approach borrows some techniques from existing algorithms (mostly direct sampling), so that part of the methodology is described in other papers. Although, some parameters are common with DS, the philosophy is quite different as DS is never explicitly calculating cpdf. In that sense, the proposed methodology is closer to classical

approaches such as snesim (except that the cpdf are not stored). I would therefore recommend to explicitly describe every part of the algorithm, without (too many) references to previous publications. Doing so, the methodology would be self-sufficient. Part of the methodology (dissimilarity metrics and MDS) is introduced in the result section and should be moved to the methodology.

We are so sorry for that we did not describe the methodology section clearly. In the revised manuscript, we have reorganized the methodology section. Some existing information but used in the following sections was moved to a new section Background Information, such as the pattern distance (see section 2.1, P5L8-17) and the existing probability aggregation formulas (see section 2.2, P5L18-26 and P6L1-11). The description of dissimilarity metrics and MDS were also move to the section Background Information (see section 2.3, P6L12-17).

3. In the parameter sensitivity, it is argued that some parameters are similar to DS, and thus the paper focuses on the new parameters. Nevertheless, although the proposed method borrow some ideas from DS, it is clear that the approach is different (In DS, you stop searching as soon as you find an occurrence with distance below the threshold, here you continue scanning to get the cpdf). Therefore, we cannot assume without checking that some parameters such as the threshold, the search neighborhood or the fraction of the TI that is investigated will have the same influence in DS and in the proposed approach. In addition, some interaction between the parameters is expected. For example, between the threshold and the maximum number of occurrence, or the number of available sections, some interactions can be expected. Indeed, if you increase the threshold, you will reach faster the maximum number of occurrence.

Thanks a lot for your insightful suggestions. The interaction between two important parameters of DS (distance threshold $t$ and fraction of training image to scan $f$) and two new parameters presented in our method (number of cross-sections $N_{cs}$ and maximum number of matched patterns from each training image $N_{max}$) has been added (see section 4.1.4 and Figure 8, P19L5-18 and P20). The results show that these two parameters have similar effects as in DS, so we did not discuss them separately. In addition, it will be repeated with the contents of Meerschman et al. (2013) in which the parameterization of DS was tested thoroughly.

4. The application example is not a real application, but another synthetic benchmark

using a real analog. Indeed, there is no verification data or specific application of the model. I would therefore merge section 3 and 4.

Because section 3 (section 4 in the new version) mainly focuses the parameterization and performance analysis of the proposed method, and section 4 (section 5 in the new version) aims to present a complete example in hydrology, so we kept this section, but changed the title to "Synthetic Example: 3-D Reconstruction of Hydrofacies" in the revised manuscript (see P26L5).

**Specific comments:**

5.  P3L6-25. You don't discuss the paper by Gueting et al. (2017) in the introduction, although it is a recent paper on the topic and you borrow some ideas later in the field example. You should reconsider this paragraph to describe with more details previous approaches and how your method is new.

    The description of the paper by Gueting et al. (2017) has been added in the introduction (see P3L32-34 and P4L1-2). In order to clearly explain the problems addressed in our paper, we added detailed descriptions of the limitations of existing methods in the revised version (see P3L25-26, P3L29-34, P4L1-2 and P4L19-23).

6.  P4L29-30. This sentence does not necessarily refer to the approach proposed by Comunian et al. (2012), but more globally, to any method assuming stationarity. It is always possible to use auxiliary variable to account for non-stationarity, as it is done by the s2Dcd approach in the application example.

    Thanks a lot for pointing it out. We have rewritten the corresponding sentences in the revised version (see P7L3-6).

7.  P5L3-8. For reconstruction algorithm, there might be a confusion between training image and hard data. Here, the sections that are used are both training images and hard data. To some extent, using the whole section as TI and the sub-sections as hard data is already a good way to locally constrain the simulations. The s2Dcd algorithm is already performing well in that sense.

    The sections that are used in our work are regarded as both training images and hard data, but it is different with s2Dcd. Obviously, s2Dcd uses a series of 2-D simulations to fill a 3-D domain, but a random simulation path containing all uninformed locations is used. The corresponding descriptions have been added in P4L19-22, P7L15-17, and P7L19-21.

8. P5-L9-10. The choice of the subsections within domains is not necessarily selecting the most local information. Indeed, this depends on the location of the node within the subgrid, some other sections might be closer, for example when close to a domain boarder. Since the approach does not store any cpdf, you could center the sub-domain on the node and the TI would change for each node. This would avoid the issue of case 2 in Figure 2, where many nodes closer to the node to simulate are actually out of the TI.

A great idea! But if we adopt this idea, the whole method proposed in this paper has to be changed. In fact, a 3-D domain is divided into several local parts according to the spatial relationship of the cross-sections, and then the MP statistics will be captured from the local surrounding sub-sections for a node to be simulated. This is a very good idea, and we will consider it in our future work.

9. P7L18-20. I am not sure it is intuitive, you might expect that the parallel sections are somehow correlated (except in case of strong non-stationarity), the multiplication of probabilities could then make more sense... and otherwise for perpendicular directions (if the field is highly anisotropic, the orthogonal directions are bringing totally different information)...

Because these two parallel sub-sections often contain similar patterns, and we just expect a larger number of samples and thus more robust pdf by uniting both of them, so we firstly use the additive method to aggregate the parallel ones, then a multiplicative method is used to combine the orthogonal pdfs. In order to depict it more clearly, the corresponding descriptions have been added in P9L15-20.

10. P8L15-18. In short, you compute the cpdf using all the neighborhood whose distances are below the threshold, right?

Yes, we compute the cpdf using patterns whose distances are below the threshold $t$. But we used a parameter $N_{max}$ to control the number of matched patterns since if this number is large enough, scanning to training images will be not necessary (see P13L1-3 and section 4.1.2).

11. P9L1-3. Multi-grid approach. The description is not clear. How do you divide the data event within several grids? I thought you were selecting all the data and previously simulated nodes within the radius. Are you only considering neighboring nodes that are on the current grid? Or you just mean that for the first grid there are

less points simulated? In practice, everything depends on the radius. Please clarify.

We are so sorry for the unclear description of the multiple-grids used in our work. In the revised version, we added the corresponding description (see P10L24-26). In fact, the neighboring nodes (hard data and previously simulated nodes) around the central node on the current grid are selected to build a data event according to the radius $R$ and the maximum number of points in the neighborhood. Therefore, a large data event is divided into several small parts placed on the different grids which results in smaller neighborhoods on each grid.

12. P9L4-5. I understand that you take the "diagonal nodes" (figure 3) for the smaller grid as they are previously simulated in the multi-grid 2, but the other nodes should also be consider (horizontally and vertically) according to your radius, thus 8 nodes in 2D and 26 nodes in 3D, no? The remark about 3D is strange since you are only considering 3 directions in 2D, this is not a 3D neighborhood or am I missing something? How do you consider previously simulated points or hard data that are out of the 2D planes? Are they simply disregarded, or somehow projected on the plane?

Yes, other nodes will be considered in other two directions. Thank you very much for pointing it out. We have deleted this sentence and added the corresponding description to make it more clear (see P10L29 and P11L2-4). In the local search strategy proposed in this work, three planes through the current simulated node in three orthogonal directions are considered. Other nodes out of the 2D planes will be disregarded (see P11L2-4).

13. P18. Figure 9. The red line shows the proportion in the cross-section, what about the "true" proportion?

The proportions of facies in the 3-D reference have been added and marked by black lines in this Figure in the revised version (see P22L4-6 and Figure 10).

14. P19L6-14. You could actually quantify how realizations are realistic using kernel smoothing (to average the information of the different realizations in a metrics), estimating the density distribution of the realizations around the reference. You can refer to the already cited paper of Hermans et al. (2015) for an application of MDS in the context of 3D MPS in hydrology.

Very useful comment! We have used kernel smoothing to estimate the density

distribution of the realizations of three different MPS approaches around the reference (see P24L1-5). Moreover, the related information about kernel smoothing has been added in section 2.3 in the new version (see P6L18-22).

15. P20L15-16. Still, in DS no cpdf is computed as the first matching sample is selected. Here, you continue scanning, so the effect of those parameters is not necessarily similar. For example, an interaction between t, the number of sections and Nmax is expected.

The interaction between $t$, $f$, $N_{cs}$ and $N_{max}$ has been added in the revised version (see section 4.1.4 and Figure 8, P19L5-18 and P20).

16. P22L6. You say that the number of sections is insufficient, but you don't provide guidelines about a sufficient density to use your algorithm. What if the sections are not oriented in orthogonal directions?

A recommended density for the number of sections has been given in P14L18-19. If the sections are not straight or orthogonal, we need to project them in orthogonal directions. The detailed description has been added in the revised manuscript (see P31L6-10 and P32Figure19).

17. P23L2-3. A short description of the auxiliary variable would be welcome. I guess it describes the proportion of facies in the different zones along the vertical direction. It does not have to be long.

A short description of the auxiliary variable has been added in revised manuscript (see P27L6-8).

18. P26L2-3. "partial lower dimensional data" is not true. You can only use 2D orthogonal sections in a sufficient amount. Borehole or analogs cannot be used since you argue that local information is important.

In order to illustrate how partial lower dimensional data are used in our work, we added a figure and the corresponding description in the revised version (see P31L6-10 and P32Figure19).

19. P26L7. But you need a lot of 2D sections, which is a clear drawback of the method.

When there are very few or no sections in a direction, a feasible solution has been suggested by Gueting et al. (2017) where sequential 2D simulations are performed to obtained some sections first, and then both the original informed data and the

obtained sections are used to reconstruct the model of the entire 3-D domain (see P14L19-21).

**Technical comments:**

20. P12L6-7. Does the "maximum search distance" correspond to the Radius of previous sections?

    Thanks a lot for pointing it out! We have used consistent description for this in the revised manuscript (see P14L6 and P21L7)

21. P21L9. You previously mentioned (P17L15) that s2Dcd was using DS with 4 processors. Please check.

    It has been corrected in the revised version (see P25L14-19).

22. P23L11. I think it should be Figure 17 instead of 16.

    Thank you for pointing it out! It has been corrected in the revised version (see P28L3-4).

**Response to Reviewer #3:**

This paper presents a locality-based MPS approach to reconstruct 3D geological models based on easily available 2D training images. To fulfil the objective, the MPS search engine roams over only several local sub-sections closer to the simulated node, instead of using a full training image. The authors also perform a parameter sensitivity analysis and the performance comparison with other previous 3D reconstruction techniques, illustrating the effectiveness of their approach using synthetic and real geological data. The results identify better performance both in portraying complex heterogeneous structures and in CPU cost.

All together it is a very good paper, well written and showing a clear and valuable contribution that deserves publication. However, a number of significant issues need to be addressed for this manuscript to be publishable. Therefore, the authors are nevertheless invited to consider carefully the following comments to improve their manuscript.

Thank you very much for your positive and constructive comments and suggestions. We have corrected all the issues you raised in the revised version. The following is a point-by-point response according to your comments.

**General comments:**

1.  I am not totally convinced with the overall contribution of this method compared to s2Dcd. This needs to be explained in detail how the proposed technique differs from s2Dcd, which is now lacking in the introductory part.

    We are so sorry for that. We have added some descriptions in the revised manuscript to explain the differences between our method and s2Dcd clearly (see P3L29-32 and P4L19-22).

2.  The MDS shows slight improvement in terms of MP simulations using the proposed scheme. The computational benefit only appears with abundant sections available in each direction, which is in practice seldom existing and also mentioned as a limitation in the manuscript. Moreover, the improvement with reproduction of non-stationary patterns might have sampling effect as only one realization is considered from each method.

    We used kernel smoothing to estimate the density distribution of the realizations of

three different MPS approaches around the reference (see P24L1-5). The result quantifies the advantages of our approach compared to DS and s2Dcd. Because 4 processors are used in DS and s2Dcd, so our method presents the speedups of about 4 compared to s2Dcd and about 120 compared to DS in this test (see P25L14-19). In addition, if there are very few or no sections in a direction, a feasible solution has been suggested by Gueting et al. (2017) where sequential 2D simulations are performed to obtained some sections first, and then both the original informed data and the obtained sections are used to reconstruct the model of the entire 3-D domain (see P14L19-21). Moreover, we drew the histograms of the four informed segments and the local models of 10 realizations for each MPS method in Figure 18c in the revised manuscript (see P31Figure18c). The result also illustrates the advantages of our approach in reproducing non-stationary patterns (see P30L2-8).

3.  Overall, I am struggled to understand the flow of the methodology section, e.g. how the multigrid concept is implemented in searching the neighborhoods, or am I missing something in the workflow of the algorithm? I would also like to see the effects of using various number of multigrid in the form of sensitivity analysis.

We are so sorry for that we did not describe the multiple-grids used in this work clearly. In the revised version, we added the corresponding description (see P10L24-26 and P11L2-4). In fact, the neighboring nodes (hard data and previously simulated nodes) around the central node on the current grid are selected to build a data event according to the radius $R$ and the maximum number of points in the neighborhood. Therefore, a large data event is divided into several small parts placed on the different grids which results in smaller neighborhoods on each grid. Moreover, the effect of the multiple-grids used in this work on computational efficiency is same as the existing ones, so we do not analyze its sensitivity. The main contribution of our strategy focuses on the ability to reproduce features with different scales. It can be observed that our method allows reproducing heterogeneous structures at different scales (see P29Figure17cd).

**Specific Comments:**

4.  P7L2-3: Rewrite the sentence.

This sentence has been rewritten in the revised manuscript (see P16L8-10).

5.  P12L12: the connectivity 'becomes'

It has been corrected in the revised version (see P14L12).

6. P12L14: I would prefer to see an example of artifacts clearly visible on a section of the reconstructed model (maybe with the example of 6x6x6 model in Figure 5), to have the feeling of how bad it is and also to justify the logic behind not using too many cross-sections.

   The first section along X direction of a reconstruction for each case has been added in Figure 5 in the revised manuscript (see P15Figure5). It can be seen that using too many cross-sections will lead to a number of artifacts.

7. P12L18: it 'is' related

   It has been corrected in the revised version (see P14L17).

8. Figure 5: Describe the black and gray lines by adding legend or in figure caption. I think the black lines represent the reference model? Also add the axes labels in variogram and connectivity plots.

   Yes, the black lines represent the corresponding features of the reference models. We have added the descriptions for the black and gray lines and the axes labels in variogram and connectivity plots in the revised manuscript (see P15).

9. P14L8-10: Rewrite the sentence as it's hard to follow in this format.

   This sentence has been rewritten in the revised manuscript (see P16L8-10).

10. P14L14: 120? or 160 or 320?

    Thanks a lot for pointing it out! It should be 160 and has been corrected in the revised version (see P16L14).

11. P17L19: analyze 'the' performance.

    Thank you for pointing it out! We have added "the" before "performance" in P21L15.

12. P17L21: our method

    It has been corrected in the revised version (see P21L17).

13. Figure 8: Caption is incomplete

    The caption of Figure 9 has been corrected in the revised version (see P22L2).

14. Figure 9: The proportions of the facies in the 3D reference could be added as well

in the plot for comparison.

The proportions of facies in the 3-D reference have been added and marked by black lines in this Figure in the revised version (see P22L4-6 and Figure 10).

15. P20L15-16: A brief summary of all other optimized parameters would be helpful for the readers.

A brief summary of all other parameters for computational efficiency has been added in the revised version (see P25L2-4).

16. Figure 13: The figure is redundant as all these numbers are already in the tables.

This figure and the corresponding description have been deleted in the revised version (see P24L14-19, P25L1-2 and P25L6-8).

17. P21L9: s2Dcd uses DS as an external MPS engine as mentioned in P17 L15-16, therefore s2Dcd also runs on 4 processors, I believe. However, the authors claimed the opposite here. Please clarify.

It has been corrected in the revised version (see P25L14-19).

18. P22L6: parts 'of' subdomains

It has been corrected in the revised version (see P26L10).

19. P23L5-6: Figure 17 compares the dissimilarity between the sections extracted from the realizations and the informed sections, and I am guessing the sections are selected as random and the authors avoid the sections those are already used as training images?

In fact, all the sections along two directions are exacted, which include both reconstructed sections and informed sections. For each realization, 70 sections (67 reconstructed sections and 3 informed sections) from xz direction and 280 sections (275 reconstructed sections and 5 informed sections) from yz direction are used to draw the MDS maps respectively. The corresponding descriptions have been added in the revised manuscript (see P27L10-14).

20. P23L11: Figure 17 instead of Figure 16.

Thank you for pointing it out! It has been corrected in the revised version (see P28L3-4).

21. P25L1: The segments in Figure 18b are chosen from three local models, so is there

any sampling effect when you select the sections to compare the reproduction of non-stationary patterns? What if you take an ensemble of sections from few realizations to compare the techniques?

Three segments are randomly selected from the three local models. We drew the histograms of the four informed segments and the local models of 10 realizations for each MPS method in Figure 18c in the revised manuscript (see P31Figure18c and P30L2-8). If the surrounding sub-sections of a local area do not contain an attribute but it exists in other locations, the patterns with this attribute will not be moved to this local area in our approach. The corresponding description has been added in P30L2-8.

[revised manuscript text omitted]

without storing patterns and without the need of multiple grids. One of the main advantages of this approach is that several types of distances between patterns can be considered, making it possible to simulate continuous variables, or even multivariate simulation. As an approximation, pattern distance was used to express the matching degree between the neighborhood of a node and a data event in the training image (*Chugunova and Hu*, 2008; *Mariethoz et al.*, 2010, 2015). ~~For the pdf-based MPS methods, using the distances between patterns greatly decreases the amount of stored patterns. Some patch-based methods (*Arpat and Caers*, 2007; *Honarkhah and Caers*, 2010; *Tahmasebi et al.*, 2012; *Zhang et al.*, 2006) were proposed on the basis of this concept. By means of computer graphics, two very efficient MPS algorithms (*Li et al.*, 2016; *Mahmud et al.*, 2014) were developed to decrease the computational burden of traditional methods.~~

[revised manuscript text omitted]

---

## Author Response (AR2)

**Response to Reviewer #2:**

The revised version of your manuscript "Locality-based 3-D multiple-point statistics reconstruction using 2-D geological cross-sections" is strongly improved compared to the initial submission. You answered thoroughly my comments and I think the paper is ready for publication. I only have a few remarks that could help to clarify some points of the paper:

We really appreciate your affirmation and the constructive comments and suggestions. We have corrected/answered all the issues you raised in this revised version. The following is a point-by-point response according to your comments.

1. Have you considered giving a name to your approach? It would help removing the occurrence in the text of "our approach" along with DS and S2Dcd.

   Thanks a lot for your suggestion. We have named our approach as 3DRCSs which is from "**3D R**econstruction using 2D **C**ross-**S**ection**s**". All the "our approach" and "our method" in the text and figures have been replaced by the new name in the revised version.

2. P3L4: remote sensing and geophysical images?

   Thank you for pointing it out. Geophysical images are often used in MPS methods as the input data. It has been corrected in the revised manuscript (see P3L4-5).

3. P8L8-20: I still don't understand the explanation of the aggregation method. Maybe I wrongly understand your sentences. You first state that log-linear pooling is used to combine probabilities with significant correlation. Then you say that two parallel sub-sections often contains similar pattern (to me, this corresponds to the idea of correlation) and that you thus use an additive aggregation operator. From my point of view, there is a contradiction here.

   We are so sorry for the unclear description of the aggregation method used in our work. In the revised version, we corrected the corresponding description (see P8L16-17). In fact, we use an additive aggregation operator to combine the two probability distributions from parallel sub-sections because we just expect a larger number of samples and thus more robust pdf by uniting both.

4. P9L26-27. If I understand correctly, the data event is not really divided on several grids. If you have many points available close to the node to simulate, there will

still be considered in the 1st multi-grid and the radius will be smaller through the maximum number of neighbours. The chance it happens is just low.

We are so sorry for that we provided a wrong figure (the bottom half of Figure 3) in the previous versions. As described in the text of the manuscript, a large data event is divided into several small parts placed on the different grids. However, all the informed and simulated nodes located in the circle with a radius $R$ are used to obtain a neighborhood in Figure 3 of the previous versions. In the revised manuscript, Figure 3 has been corrected (see P10 Figure 3) and only the informed and simulated nodes on the current grid are considered.

5. P10L1-3. I would explicitly state that out of plane hard data will be disregarded.

It is true. Only the informed nodes on three planes through the current simulated node in three orthogonal directions are considered to obtain the neighborhoods in our approach. This is because there are only 2-D cross-sections to be scanned and no 3-D training image. Although the other hard data nearby the current node are disregarded for the simulation of this node, they will definitely be used in the simulation of other nodes.

6. P10. Algorithm 1 - Point 4. Your process is fully random, i.e. that the random path is in the 3D space and not within a specific section (from what I understand). I think it is worth mentioning.

Thanks a lot for pointing it out. Exactly, we use a fully random path on each multiple grid in the 3-D space. We have added the corresponding description in the revised version (see P11L3-6).

7. P15. Table 2. It is probably out of the scope of the paper, but it seems that the number of geobodies cannot be reproduced using directional 2D images. Do you think it would be better with a 3D image or is it rather related to the stochastic simulation process?

Thank you for pointing this out. Although the number of geobodies is not reproduced well as shown in Table 2, it can still be used as a metric to evaluate the sensitivity of the parameters because the number of geobodies varies significantly when the corresponding parameters change. We used DS to perform a MPS-based reconstruction using 3*3*3 cross-sections and a MPS simulation using a complete 3D training image. We found that the number of geobodies was reproduced better

when using a 3D training image. On the other hand, a distance threshold $t$ is adopted in both DS and our approach. This parameter may bring some noises because the mismatched patterns are accepted. Moreover, a post-processing operation was not performed in the test presented in this manuscript.

8. P22L11-13. I guess you would get a stronger discrimination if the width of the Kernel was reduced. I am a little bit surprised that DS still gets 29% of the contribution.

We recalculated the density distribution of the realizations of three different MPS approaches around the reference by using Kernel smoothing. But the result was same with the values described in the last version.

[revised manuscript text omitted]